# Learning Expandable and Adaptable Representations for Continual Learning

**Ruilong Yu[1], Mingyan Liu[2], Fei Ye[1],[*] Adrian G. Bors[3],**
**Rongyao Hu[1], Jingling Sun[1], and Shijie Zhou[1]**
[1]University of Electronic Science and Technology of China, Chengdu, China
[2]Harbin Institute of Technology, Shenzhen, China
[3]University of York, York, U.K.
`yrl666@outlook.com`, `2023312903@stu.hit.edu.cn`, `feiye@uestc.edu.cn`,
`adrian.bors@york.ac.uk`, `ryhu@uestc.edu.cn`,
`jingling.sun910@gmail.com`, `sjzhou@uestc.edu.cn`

## Abstract

Extant studies predominantly address catastrophic forgetting within a simplified continual learning paradigm, typically confined to a singular data domain. Conversely, real-world applications frequently encompass multiple, evolving data domains, wherein models often struggle to retain many critical past information, thereby leading to performance degradation. This paper addresses this complex scenario by introducing a novel dynamic expansion approach called Learning Expandable and Adaptable Representations (LEAR). This framework orchestrates a collaborative backbone structure, comprising global and local backbones, designed to capture both general and task-specific representations. Leveraging this collaborative backbone, the proposed framework dynamically creates a lightweight expert to delineate decision boundaries for each novel task, thereby facilitating the prediction process. To enhance new task learning, we introduce a novel Mutual Information-Based Prediction Alignment approach, which incrementally optimizes the global backbone via a mutual information metric, ensuring consistency in the prediction patterns of historical experts throughout the optimization phase. To mitigate network forgetting, we propose a Kullback–Leibler (KL) Divergence-Based Feature Alignment approach, which employs a probabilistic distance measure to prevent significant shifts in critical local representations. Furthermore, we introduce a novel Hilbert-Schmidt Independence Criterion (HSIC)-Based Collaborative Optimization approach, which encourages the local and global backbones to capture distinct semantic information in a collaborative manner, thereby mitigating information redundancy and enhancing model performance. Moreover, to accelerate new task learning, we propose a novel Expert Selection Mechanism that automatically identifies the most relevant expert based on data characteristics. This selected expert is then utilized to initialize a new expert, thereby fostering positive knowledge transfer. This approach also enables expert selection during the testing phase without requiring any task information. Empirical results demonstrate that the proposed framework achieves state-of-the-art performance. Code is available at `https://github.com/yrluestc/NeurIPS2025-LEAR`.

---

[*]Corresponding author: feiye@uestc.edu.cn.

39th Conference on Neural Information Processing Systems (NeurIPS 2025).

# 1 Introduction

Modern deep learning frameworks have shown exceptional effectiveness across various visual tasks [15, 19]. However, the high performance of these approaches largely depends on large datasets, which are frequently unavailable in settings marked by constant change. This approach to learning is known as Continual/Lifelong Learning (CL), which aims to create a model that can continuously integrate new information while preserving all previously learned knowledge. Catastrophic forgetting is a significant issue that hinders the model's performance on earlier tasks [32], arising when the model tries to adjust its parameters to learn new tasks.

Among the various approaches to mitigate catastrophic forgetting in continual learning [41], Expansion-Based Methods (EBMs) have emerged as leading and highly effective strategies. The core idea is to dynamically expand the model's internal structure by adding task-specific modules to allocate dedicated capacity for each new task. However, current EBMs primarily focus on Class-Incremental Learning (CIL) [37] within a single domain, neglecting the scenario of learning across multiple domains, known as Domain-Incremental Learning (DIL). Although studies [44, 51, 21] have investigated DIL, their evaluated domains (e.g., Aircraft [29], MNIST [27]) have achieved near-perfect accuracy with pre-trained ViTs [10], making these benchmarks inadequate for assessing genuine continual learning capabilities. Therefore, we establish a more challenging and more realistic Multi-domain Continual learning (MDCL) scenario, where the task sequence comprises not only complex domains with large discrepancies but also a mixture of domains with underlying similarities. In this study, we aim to improve the model's performance in MDCL by considering three aspects including plasticity, stability and efficiency. To implement this goal, we propose a novel approach called LEAR and its core idea is to fully explore the stable and dynamic representations extracted by the pre-trained ViT backbones to achieve fast adaptation while adaptively optimizing the backbones to maintain all previously learned information.

(1) Plasticity. Existing EBMs improve downstream task performance by integrating task-specific prompts [46, 36] or adapters [30, 49] into a fixed pretrained backbone. However, these methods focus on exploring representations from a single pre-trained backbone, which fails to address more challenging data domains such as CUB-200 [39] and ImageNet-R [17]. Thus, to improve plasticity in a challenging MDCL scenario, we introduce a novel collaborative backbone architecture for LEAR, comprising a global and a local backbone, designed to capture general and task-specific information across all tasks. Leveraging this collaborative backbone structure, the proposed LEAR framework dynamically generates a lightweight expert to learn the decision boundary for each new task, thereby achieving commendable performance. The results presented in Tab. 1 and 2 demonstrate that our method achieves superior performance on most individual datasets in the MDCL scenario, which also validates that EBMs with frozen pretrained backbones cannot provide sufficient plasticity in MDCL.

(2) Stability. Many EBMs have been shown to achieve excellent stability in CIL. However, the excellent stability is usually achieved by freezing all parameters of the pre-trained models during the training, which may lead to forgetting of historical tasks in MDCL, especially when facing the severe domain shifts (e.g. ChestX [43] $\rightarrow$ ImageNet-R) in long task sequences. To address this limitation, we propose a unified optimization function to regulate the optimization behaviour of the collaborative backbone structure. This function consists of a Mutual Information-Based Prediction Alignment (MIBPA) loss and a Kullback–Leibler Divergence-Based Feature Alignment (KLDBFA) loss. The former dynamically optimizes the global backbone while preventing negative knowledge transfer at the prediction level, and the latter aligns historical and current representation distributions at the feature level. Such a design enables LEAR to achieve rehearsal-free continual learning by actively consolidating historical knowledge at both the prediction and feature levels when fine-tuning the collaborative backbones with new task data, rather than freezing parameters passively. Such a design has not been explored in the existing CL field. Furthermore, to mitigate optimization interference and information redundancy between the collaborative backbones, we propose a novel Hilbert-Schmidt Independence Criterion-Based Collaborative Optimization (HSICBCO) strategy to encourage two backbones to capture different semantic information, thus promoting effective complementary learning of MDCL tasks. The experimental results demonstrate that LEAR significantly outperforms all baseline methods in terms of overall average accuracy in three MDCL scenarios.

(3) Efficiency. Many existing EBMs usually ignore the task relevance and do not explore the previously learned parameter information to accelerate the new task learning. As a result, these methods optimize each new expert from scratch, which may result in considerable computational

costs and parameter redundancy when dealing with MDCL that contains analogous data domains. To address this issue, we aim to promote the efficient learning process of LEAR by proposing a novel Expert Selection Mechanism (ESM) that selectively transfers the parameter information learned by a selected expert into the new expert construction process. Specifically, the proposed ESM models each expert's knowledge as a Gaussian memory distribution and only preserve its critical statistical information. For each new task, the proposed ESM selects the most relevant expert by minimizing the Mahalanobis distance between stored distributions and incoming data, and reuses its parameters to facilitate new task learning. During the testing phase, ESM autonomously routes testing samples to the most suitable expert in a task-agnostic manner.

The principal contributions of this research are enumerated as follows : (1) This paper explores the challenging MDCL scenarios by proposing a novel approach called Learning Expandable and Adaptable Representations (LEAR) that optimizes and manages a collaborative backbone structure, comprising a global backbone and a local backbone, respectively. This design can help capture general and task-specific representations, which achieve excellent performance in MDCL. (2) A novel MIBPA approach is proposed to optimize the global backbone via a mutual information measure that ensures the consistency of the prediction pattern of each history expert when adjusting the parameters of the global backbone. (3) A novel KLDBFA approach is proposed to regulate the optimization behaviour of the local backbone by preventing significant changes in many critical local representations. Such a design can preserve task-specific representation information and prevent significant negative knowledge transfer effects. (4) A novel HSIBCO strategy is proposed to enforce the disentanglement between global and local representations, which avoids information redundancy and improves the model's performance. (5) A novel ESM is proposed to select the most relevant expert according to the data's characteristics, which is used in the training phase to promote the positive knowledge transfer process and in the testing phase to implement the expert selection procedure. The results from an extensive suite of experiments demonstrate that our proposed approach significantly outperforms existing baselines across all experimental configurations.

## 2 Related Work

**Rehearsal-based techniques** represent a widely adopted strategy for mitigating forgetting by dynamically incorporating a limited number of historical examples into the memory buffer [5, 6]. These memory samples are leveraged alongside new training instances to enhance model performance during the new task learning. Thus, the quality of the memorized samples is paramount within the rehearsal-based optimization framework [14]. Moreover, rehearsal-based approaches can be augmented through the integration of regularization techniques, with the objective of further elevating the overall efficacy of the model [2, 9, 20, 45]. In addition, memory studies have proposed to train the generative models to implement the memory system, which can provide infinite generative replay samples [1, 33, 35, 50, 23].

**Prompt-based techniques** leverage frozen pre-trained models like Vision Transformers (ViT) [10] as feature extractors, adapting them to sequential tasks through task-specific learnable prompt parameters. Current approaches employ diverse prompt management strategies including L2P [47]'s shared prompt pool with query-key retrieval mechanism, DualPrompt [46]'s separation of task-agnostic (G-Prompt) and task-specific (E-Prompt) components, and CODA-Prompt [36]'s attention-weighted cross-task prompt expansion. HiDe-Prompt [40] further advances performance by hierarchically decomposing class-incremental learning objectives for optimized task adaptation.

**Expansion-based methods** represent a robust approach to mitigating network forgetting in continual learning [8]. Such an approach dynamically expands the network architecture to enhance the learning ability of the new task [22, 38]. Beyond convolutional neural networks, expansion-based techniques have also been explored to leverage the capabilities of ViTs as the foundational network. These methods usually create self-attention blocks with the task-specific classifier to adapt to the new task learning [11, 48, 30, 49]. However, these methodologies typically involve freezing the pre-trained model, which limits their adaptability to complex and unknown data domains. We provide additional information on the related work in **Appendix-A** from Supplementary Materials (SM).

## 3 Methodology

### 3.1 Problem Definition

CL seeks to develop a model capable of acquiring knowledge across multiple sequences of tasks while retaining previously acquired information. This paper addresses a more pragmatic learning context in

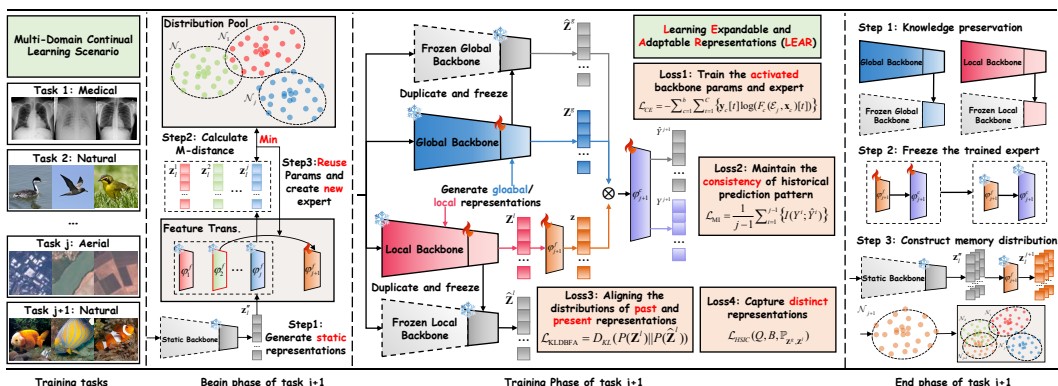

Figure 1: The training framework of the proposed LEAR. The data samples from new tasks are processed through a collaborative backbone structure to learn task-shared, task-specific and backbone-distinct representations via the proposed MIBPA, KLDBFA and HSICBCO, respectively. ESM constructs memory distributions and selects relevant experts for network expansion and test evaluation.

which each task encompasses previously unseen challenging data domains. Let $\mathcal{D}_i^S = \{\mathbf{x}_j, \mathbf{y}_j\}_{j=1}^{n_i^S}$ and $\mathcal{D}_i^T = \{\mathbf{x}_j, \mathbf{y}_j\}_{j=1}^{n_i^T}$ denote the $i$-th training and testing datasets, respectively. In a class-incremental learning scenario [3], a data split procedure is performed to divide the training dataset $\mathcal{D}_i^S$ into $C_i$ subsets $\{\mathcal{D}_{i,1}^S, \cdots, \mathcal{D}_{i,C_i}^S\}$ according to the category, where each task $\mathcal{T}_j$ is associated with a training dataset $\mathcal{D}_{i,j}^S$ formed by samples from several adjacent classes. In the context of a specific task learning $\mathcal{T}_j$, the model is restricted to utilizing only the training dataset $\mathcal{D}_{i,j}^S$, with all preceding datasets $\{\mathcal{D}_{i,1}^S, \cdots, \mathcal{D}_{i,j-1}^S\}$ being inaccessible. In DIL, each task is conceptualized as a distinct data domain, denoted as $\{\mathcal{D}_1^S, \cdots, \mathcal{D}_n^S\}$, with $n$ representing the total number of tasks. In contrast to these two CL scenarios which posit that each task comprises non-overlapping and heterogeneous data samples, a new task within our MDCL encompasses data samples that exhibit not only similar semantic characteristics with previously seen tasks but also significant domain shifts. Consequently, it becomes imperative to leverage existing parameters to facilitate learning these analogous tasks, thereby accelerating the training process and minimizing resource consumption. Once the final task learning is finished, the model's efficacy is assessed across all testing datasets $\{\mathcal{D}_1^T, \cdots, \mathcal{D}_n^T\}$ through the lens of classification accuracy.

## 3.2 Collaborative Backbone Structure

Recent investigations in CL have assessed the efficacy of leveraging a pre-trained ViT [10] to enhance model performance. These methodologies typically incorporate the pre-trained ViT as the primary backbone, facilitating the expert construction process while concurrently freezing its parameters to mitigate catastrophic forgetting. Nevertheless, this architectural design constrains the model's capacity for learning in the context of novel tasks, particularly when the incoming data exhibits divergent domain characteristics. This paper addresses this limitation by introducing a novel collaborative backbone architecture, comprising global and local backbones, each instantiated via a pre-trained ViT to facilitate rapid adaptation. Specifically, the global backbone incrementally updates its parameters throughout the optimization phase, with the objective of generating a task-shared representation applicable across tasks. Conversely, the local backbone is engineered to dynamically adapt to new tasks through parameter adjustments. We propose to optimize only the final three layers of the global and local backbone to mitigate computational expenses.

Let $F_{\theta^g} : \mathcal{X} \to \mathcal{Z}'$ denote a global backbone, implemented using a pre-trained ViT, which receives a data sample $\mathbf{x}$ over the data space and returns a feature vector $\mathbf{z}'$ over the feature space $\mathcal{Z}'$. Similarly, let $F_{\theta^l} : \mathcal{X} \to \mathcal{Z}'$ denote a local backbone, which has the same input-output pattern as the global backbone. For a given data sample $\mathbf{x}$, we can obtain its feature representations extracted by the global and local backbones, expressed as :

$$\mathbf{z}_g = F_{\theta^g}(\mathbf{x}), \mathbf{z}_l = F_{\theta^l}(\mathbf{x}), \tag{1}$$

By using Eq. (1), the proposed framework dynamically creates a lightweight expert ($\mathcal{E}_j$) consisting of a simple feature transformation module $F_{\varphi_j^f} : \mathcal{Z}' \to \mathcal{Z}$ and a linear classifier $F_{\varphi_j^c} : \mathcal{Z} \to \mathcal{Y}$, aiming to learn a decision boundary for a specific task. $F_{\varphi_j^f}$ receives the local representation $\mathbf{z}_l$ and returns a feature vector $\mathbf{z}$ over the feature space $\mathcal{Z}$, which is concatenate with global representation $\mathbf{z}_g$ and fed into the linear classifier $F_{\varphi_j^c}$ to make the prediction over the space $\mathcal{Y}$. The subscript $j$ denotes the expert index, and $\oplus$ denotes the concatenation operation that combines two representations into a single feature vector. The prediction process of the $j$-th expert is expressed as :

$$F_c(\mathcal{E}_j, \mathbf{x}) = F_{\varphi_j^c}(F_{\theta^g}(\mathbf{x}) \oplus F_{\varphi_j^f}(F_{\theta^l}(\mathbf{x}))). \tag{2}$$

By integrating representations derived by global and local backbones, the expert $\mathcal{E}_j$ in Eq. (2) can improve its generalization performance for a given data sample $\mathbf{x}$.

### 3.3 Mutual Information-Based Prediction Alignment

The global backbone's objective is to furnish a unified representation across all observed tasks. Consequently, optimization of the global backbone is susceptible to catastrophic forgetting, impacting all historical experts. To mitigate this, we introduce a novel Mutual Information-Based Prediction Alignment (MIBPA) methodology, designed to maintain the consistency of predictions of all historical experts when changing the parameters of the global backbone during the acquisition of new tasks. Specifically, we construct a parameter-shared auxiliary model $F_{\hat{\theta}^g}$ by replicating and freezing the global backbone's final three layers, then connecting them in parallel with intermediate features from the backbone's preceding layers. This auxiliary model subsequently guides the global backbone's optimization, producing two distinct prediction sets through the i-th expert, formulated as: :

$$\begin{aligned} \mathbf{Y}^i &= \left\{ \mathbf{y}_c \,|\, \mathbf{y}_c = F_{\varphi_i^c}(F_{\theta^g}(\mathbf{x}_c) \oplus F_{\varphi_i^f}(F_{\theta^l}(\mathbf{x}_c))), c = 1, \cdots, b \right\}, \\ \hat{\mathbf{Y}}^i &= \left\{ \mathbf{y}_c \,|\, \mathbf{y}_c = F_{\varphi_i^c}(F_{\hat{\theta}^g}(\mathbf{x}_c) \oplus F_{\varphi_i^f}(F_{\theta^l}(\mathbf{x}_c))), c = 1, \cdots, b \right\}, \end{aligned} \tag{3}$$

where $b$ denotes the size of the data batch $\mathbf{X} = \{\mathbf{x}_1, \cdots, \mathbf{x}_b\}$ and $\mathbf{x}_c$ denotes the $c$-th data sample of $\mathbf{X}$. Let $P(Y^i, \hat{Y}^i)$ denote a joint distribution, where $P(Y^i)$ and $P(\hat{Y}^i)$ represent the marginal distributions of $\mathbf{Y}^i$ and $\hat{\mathbf{Y}}^i$, respectively. Let $Y^i$ and $\hat{Y}^i$ denote two random variables over the joint distribution $P(Y^i, \hat{Y}^i)$. The proposed MIBPA approach minimizes the mutual information between $Y^i$ and $\hat{Y}^i$, expressed as :

$$I(Y^i; \hat{Y}^i) = \sum_{\hat{\mathbf{y}}^i \in \hat{Y}^i} \left\{ \sum_{\mathbf{y}^i \in Y^i} \left\{ P(Y^i, \hat{Y}^i)(\mathbf{y}^i, \hat{\mathbf{y}}^i) \log \frac{P(Y^i, \hat{Y}^i)(\mathbf{y}^i, \hat{\mathbf{y}}^i)}{p(Y^i)(\mathbf{y}^i)p(\hat{Y})(\hat{\mathbf{y}}^i)} \right\} \right\}, \tag{4}$$

where $P(Y^i, \hat{Y}^i)(\mathbf{y}^i, \hat{\mathbf{y}}^i)$ signifies the probability density function of $P(Y^i, \hat{Y}^i)$. The mutual information term $I(Y^i; \hat{Y}^i)$, as defined in Eq. (4), evaluates the distance of the prediction made by the $i$-th expert built on the previously and currently learned global backbones. A small mutual information term $I(Y^i; \hat{Y}^i)$ indicates that updating the global backbone can still maintain the prediction pattern of the $i$-th expert. Finally, the final MIBPA regularization loss function at the $j$-th task learning is defined as :

$$\mathcal{L}_{\mathrm{MI}} = \frac{1}{j-1} \sum_{i=1}^{j-1} \left\{ I(Y^i; \hat{Y}^i) \right\}. \tag{5}$$

### 3.4 Kullback–Leibler (KL) Divergence-Based Feature Alignment

The iterative updating of the pre-trained backbones facilitates the temporal capture of local representations, thereby potentially enhancing the acquisition of novel tasks. However, this process risks inducing adverse knowledge transfer and performance degradation across historical experts. Regularization methods like EWC [24] and MAS [4], which typically impose constraints on parameter updates, are not desirable to capture the complex distributional shifts across domains, while knowledge distillation methods like LWF [28] and iCaRL [34] require maintaining additional teacher networks that become computationally prohibitive as the number of tasks grows. To address these limitations, we propose Kullback-Leibler Divergence-Based Feature Alignment (KLDBFA), designed to preserve crucial historical parameters during the optimization of the local backbone.

Our design rationale for selecting KL divergence stems from two key considerations: Firstly, modern generative evaluation metrics (e.g., FID [18], Kernel MMD [13]) operate on the Gaussian distribution assumption in high-dimensional feature spaces. This motivates us to model backbone features as Gaussian distributions. Fig. 1 in Appendix-C from the SM also provides additional empirical validation for the Gaussian distribution assumption. Secondly, KL divergence offers unique advantages over alternative distributional metrics: (1) Its directional property enables targeted constraint of current features toward historical distributions, unlike symmetric metrics (e.g., Jensen-Shannon divergence); (2) It maintains computational efficiency compared to expensive metrics like MMD or Wasserstein distance. These characteristics make KL divergence ideally suited for continual learning scenarios requiring efficient knowledge preservation.

Specifically, upon each task transition, the proposed KLDBFA approach duplicates and immobilizes the local backbone $F_{\theta^l}$ as a frozen model $F_{\hat{\theta}^l}$ following a similar procedure in MIBPA. $F_{\hat{\theta}^l}$ serves to regulate the optimization dynamics of the local backbone. For a given data batch $\mathbf{X} = \{\mathbf{x}_1, \cdots, \mathbf{x}_b\}$, two distinct sets of feature vectors are derived utilizing $F_{\theta^l}$ and $F_{\hat{\theta}^l}$, respectively, as follows :

$$\mathbf{Z}^l = \{\mathbf{z}_c \,|\, \mathbf{z}_c = F_{\theta^l}(\mathbf{x}_c), c = 1, \cdots, b\}, \hat{\mathbf{Z}}^l = \{\mathbf{z}_c \,|\, \mathbf{z}_c = F_{\hat{\theta}^l}(\mathbf{x}_c), c = 1, \cdots, b\}. \tag{6}$$

Building upon the Gaussian assumption stated above, we model two Gaussian distributions $P(\mathbf{Z}^l) = \mathcal{N}(\boldsymbol{\mu}_1, \boldsymbol{\Sigma}_1)$ and $P(\hat{\mathbf{Z}}^l) = \mathcal{N}(\boldsymbol{\mu}_2, \boldsymbol{\Sigma}_2)$, through calculating the mean vectors $\boldsymbol{\mu}_1$, $\boldsymbol{\mu}_2$ and covariance matrix $\boldsymbol{\Sigma}_1$, $\boldsymbol{\Sigma}_2$ of $\mathbf{Z}^l$ and $\hat{\mathbf{Z}}^l$, respectively. We propose to employ the KL divergence to evaluate the discrepancy between $P(\mathbf{Z}^l)$ and $P(\hat{\mathbf{Z}}^l)$ as a regularization loss term, expressed as :

$$D_{KL}(P(\mathbf{Z}^l)\|P(\hat{\mathbf{Z}}^l)) = \frac{1}{2}\left[\log\left(\frac{\det(\Sigma_2)}{\det(\Sigma_1)}\right) - d + \mathrm{tr}(\Sigma_2^{-1}\Sigma_1) + (\mu_2 - \mu_1)^\top \Sigma_2^{-1}(\mu_2 - \mu_1)\right]$$
$$\mathcal{L}_{\text{KLDBFA}} = D_{KL}(P(\mathbf{Z}^l) \,\|\, P(\hat{\mathbf{Z}}^l)),$$
$$\tag{7}$$

where $D_{KL}$ denotes the KL divergence. $\det(\cdot)$, $d$, and $\mathrm{tr}(\cdot)$ represent the determinant, dimension, and trace of a matrix, respectively.

### 3.5 HSIC-Based Collaborative Optimization

The global and local backbones are designed to capture distinct feature representations, thereby potentially improving model efficacy. To further facilitate the disentanglement between these backbones, we introduce a novel Hilbert-Schmidt Independence Criterion (HSIC)-Based Collaborative Optimization (HSICBCO) methodology. This approach leverages an independence criterion to maximize the divergence of knowledge between the global and local backbones. Specifically, we employ the HSIC measure [12], given its property of ranging from 0 to infinity, with 0 signifying statistical independence. Consequently, minimizing the HSIC term allows for enhanced disentanglement between the global and local backbones, which can be easily added to the primary loss function.

Let $Z^g$ and $Z^l$ denote two distinct domains, and let $\mathbb{P}_{\mathbf{Z}^g, \mathbf{Z}^l}$ represent a joint distribution from which a sample pair $\{\mathbf{z}_g, \mathbf{z}_l\}$ is drawn using global and local backbones across $Z^g \times Z^l$. The primary objective of HSIC, as delineated in [12], within the framework of Reproducing Kernel Hilbert Space (RKHS), [42], is to quantify the dependency between the domains of the variables $\mathbf{z}_g$ and $\mathbf{z}_l$ by assessing the norm of the cross-covariance operator over the domain $Z^g \times Z^l$. Let $Q$ and $B$ be the RKHSs defined on $Z^g$ and $Z^l$, respectively, and let $f_Q \colon Z^g \to Q$, and $f_B \colon Z^l \to B$ denote their respective feature mappings. The associated reproducing kernels are defined as $k(\mathbf{z}_g, \mathbf{z}_g') = \langle f_Q(\mathbf{z}_g), f_Q(\mathbf{z}_g')\rangle$ and $l(\mathbf{z}_l, \mathbf{z}_l') = \langle f_B(\mathbf{z}_l), f_B(\mathbf{z}_l')\rangle$, where $\mathbf{z}_g, \mathbf{z}_g' \in Z^g$ and $\mathbf{z}_l, \mathbf{z}_l' \in Z^l$. The cross-covariance operator between $f_Q$ and $f_B$ is defined as follows :

$$C_{\mathbf{z}_g \mathbf{z}_l} = \mathbb{E}_{\mathbf{z}_g \mathbf{z}_l}\left\{ \left(f_Q(\mathbf{z}_g) - \mathbb{E}_{\mathbf{z}_g}\left[f_Q(\mathbf{z}_g)\right]\right) \otimes \left(f_B(\mathbf{z}_l) - \mathbb{E}_{\mathbf{z}_l}\left[f_B(\mathbf{z}_l)\right]\right) \right\}, \tag{8}$$

where $\otimes$ is the tensor product. HSIC is defined as the square of the Hilbert-Schmidt norm of $C_{\mathbf{z}_g, \mathbf{z}_l}$ :

$$\mathcal{L}_{HSIC}(Q, B, \mathbb{P}_{\mathbf{Z}^g, \mathbf{Z}^l}) = \left\|C_{\mathbf{z}_g, \mathbf{z}_l}\right\|_{HS}^2 = \mathbb{E}_{\mathbf{z}_g, \mathbf{z}_g', \mathbf{z}_l, \mathbf{z}_l'}[k(\mathbf{z}_g, \mathbf{z}_g')l(\mathbf{z}_l, \mathbf{z}_l')]$$
$$+ \mathbb{E}_{\mathbf{z}_g, \mathbf{z}_g'}[k(\mathbf{z}_g, \mathbf{z}_g')]\mathbb{E}_{\mathbf{z}_l, \mathbf{z}_l'}[l(\mathbf{z}_l, \mathbf{z}_l')] - 2\mathbb{E}_{\mathbf{z}_g, \mathbf{z}_l}[\mathbb{E}_{\mathbf{z}_g'}[k(\mathbf{z}_g, \mathbf{z}_g')]\mathbb{E}_{\mathbf{z}_l'}[l(\mathbf{z}_l, \mathbf{z}_l')]], \tag{9}$$

where $\mathbb{E}_{\mathbf{z}_g, \mathbf{z}_g', \mathbf{z}_l, \mathbf{z}_l'}$ represents the expectation over samples $\{\mathbf{z}_g, \mathbf{z}_l\}$ and $\{\mathbf{z}_g', \mathbf{z}_l'\}$ drawn from $\mathbb{P}_{\mathbf{Z}^g, \mathbf{Z}^l}$.

## 3.6 Expert Selection Mechanism

In scenarios where analogous data domains are encountered in subsequent tasks, the reuse of pertinent parameters and information becomes imperative for the efficient learning of such domains, thereby accelerating the training process of a novel task. To this end, this study introduces a novel Expert Selection Mechanism (ESM), designed to identify the most relevant expert for a given new task, facilitating the reuse of existing parameters to initialize a new expert, which, in turn, can engender positive knowledge transfer effects.

**The memory distribution.** Specifically, we utilize a frozen pre-trained ViT as a feature extractor, denoted as $F_{\theta^f} : \mathcal{X} \to \mathcal{Z}''$, to generate static data representations, where $\theta^f$ represents the fixed parameters of the ViT backbone. To mitigate parameter redundancy, we duplicate and freeze the local backbone's final three layers as in KLDBFA in the first task. Upon the completion of a specific task learning phase ($\mathcal{T}_j$), a subset of training samples $\{\mathbf{x}_k\}_{k=1}^m$ is randomly selected from $\mathcal{D}_j^S$ and processed by $F_{\theta^f}$ to extract the class token representation $\mathbf{z}_k'' = F_{\theta^f}(\mathbf{x}_k)$. These extracted features are subsequently propagated through the fully connected layer of the current expert $\mathcal{E}_j$, yielding transformed features :

$$\mathbf{z}_k = F_{\varphi_j^f}(\mathbf{z}_k'') . \tag{10}$$

By using the transformed features, we obtain the empirical mean $\boldsymbol{\mu}_j$ and covariance matrix $\boldsymbol{\Sigma}_j$ by :

$$\boldsymbol{\mu}_j = \frac{1}{m} \sum\nolimits_{k=1}^m \{\mathbf{z}_k\}, \quad \boldsymbol{\Sigma}_j = \frac{1}{m-1} \sum\nolimits_{k=1}^m \left\{ (\mathbf{z}_k - \boldsymbol{\mu}_j)(\mathbf{z}_k - \boldsymbol{\mu}_j)^\top \right\} . \tag{11}$$

Subsequently, a multivariate Gaussian distribution $\mathcal{N}_j = \mathcal{N}(\boldsymbol{\mu}_j, \boldsymbol{\Sigma}_j)$ is constructed to preserve the statistical information about the $j$-th task. We call $\mathcal{N}_j$ as the memory distribution for the expert $\mathcal{E}_j$, which is always fixed during the subsequent learning.

**The expert selection process.** When a new task $\mathcal{T}_{j+1}$ begins, its training samples $\{\mathbf{x}_l\}_{l=1}^{m'}$ are first processed by the frozen backbone $F_{\theta^f}$ to obtain $\mathbf{z}_l'' = F_{\theta^f}(\mathbf{x}_l)$. For the $l$-th representation $\mathbf{z}_l''$, we can employ the feature transformation modules $\{F_{\varphi_1^f}, \cdots, F_{\varphi_j^f}\}$ of all existing experts $\{\mathcal{E}_1, \cdots, \mathcal{E}_j\}$ to generate a set of transformed features $\mathbf{z}_l^c = F_{\varphi_j^f}(\mathbf{z}_l''), \quad \forall c = 1, \cdots, j$. Based on the transformed features, the most relevant expert $\mathcal{E}_{c^*}$ with the minimum average Mahalanobis distance at the new task learning ($\mathcal{T}_{j+1}$) is selected by :

$$c^* = \underset{c=1,\cdots,j}{\operatorname{argmin}} \left\{ \frac{1}{m'} \sum\nolimits_{l=1}^{m'} \sqrt{(\mathbf{z}_l^c - \boldsymbol{\mu}_c)^\top \boldsymbol{\Sigma}_c^{-1} (\mathbf{z}_l^c - \boldsymbol{\mu}_c)} \right\} , \tag{12}$$

where $c^*$ denotes the index of the selected expert. Finally, a new expert $\mathcal{E}_{j+1}$ is initialized using the parameters of the selected expert $\mathcal{E}_{c^*}$, particularly inheriting its feature transformation module $F_{\varphi_{c^*}^f}$ and linear classifier $F_{\varphi_{c^*}^c}$. The new expert is then fine-tuned on the subset of incoming task samples to adapt to the new domain. During inference, test samples are assigned to the most suitable expert for prediction using the same selection strategy. The reason for choosing the Mahalanobis distance is that it accounts for the scale and correlation of variables, making it suitable for datasets where features are correlated or have different units. Other distance measures are analyzed in **Appendix-D** from SM. This Mahalanobis distance-based expert selection strategy enables the model to dynamically identify the most semantically related expert for knowledge transfer, thereby accelerating convergence and reducing parameter overhead during continual learning.

## 3.7 Algorithm Implementation

The learning procedure of the proposed LEAR (illustrated in Fig. 1) consists of three stages:

**Step 1: Collaborative backbone initialization.** We initialize both global and local backbones using pre-trained ViT models. These networks serve as the feature extractors for all tasks throughout the Multi-Domain Continual Learning process. For a given input $\mathbf{x}$, we obtain its feature representations extracted by both backbones using Eq. (1).

**Step 2: Dynamic expert creation and selection.** During the training of the $j$-th task ($\mathcal{T}_j$), we dynamically create a lightweight expert tailored to this domain. To ensure effective knowledge transfer, we select the most relevant historical expert based on the Mahalanobis distance computed

Table 1: The classification accuracy (%) of all testing datasets after learning the **CDM** task sequence.

| Methods | C10 | Disease | MNIST | RESISC45 | EuroSAT | TImg | C100 | ChestX | ImgR | CUB200 | Avg |
|---|---|---|---|---|---|---|---|---|---|---|---|
| DER++(Re) | 22.78 | 34.00 | 11.33 | 8.24 | 18.18 | 7.66 | 31.79 | 25.99 | 54.04 | 78.23 | 29.22 |
| CLS-ER | 22.58 | 27.22 | 12.50 | 14.32 | 24.17 | 14.14 | 34.23 | 25.99 | 52.46 | 75.78 | 30.34 |
| RanPAC | 87.17 | 96.76 | 87.45 | 84.20 | 92.53 | 71.46 | 51.44 | 39.63 | 43.30 | 56.18 | 71.01 |
| MoE | 92.74 | 32.44 | 91.87 | 54.02 | 41.85 | 6.12 | 78.98 | 19.60 | **78.43** | 77.24 | 57.33 |
| L2P | 20.66 | 11.15 | 14.08 | 4.50 | 13.65 | 11.67 | 31.76 | 21.52 | 58.26 | 81.30 | 26.85 |
| DAP | 8.83 | 2.78 | 18.94 | 3.36 | 18.19 | 3.06 | 11.29 | 16.34 | 60.68 | 80.37 | 22.38 |
| D-Prompt | 25.99 | 9.08 | 16.57 | 4.13 | 7.30 | 22.73 | 38.36 | 17.33 | 59.06 | 82.44 | 28.30 |
| C-Prompt | 13.57 | 2.33 | 9.10 | 1.90 | 14.18 | 0.68 | 3.40 | 14.35 | 4.14 | 60.55 | 12.42 |
| Ours | **95.44** | **98.46** | **96.59** | **92.04** | **95.00** | **81.24** | **85.10** | **45.95** | 70.47 | **85.80** | **84.61** |

Table 2: The classification accuracy (%) of all testing datasets after learning the **ETI** task sequence.

| Methods | EuroSAT | TImg | ImgR | CUB200 | C100 | MNIST | RESISC45 | ChestX | C10 | Disease | Avg |
|---|---|---|---|---|---|---|---|---|---|---|---|
| DER++(Re) | 51.82 | 36.25 | 2.32 | 6.20 | 23.08 | 65.97 | 40.45 | 27.41 | 82.69 | 97.77 | 43.40 |
| CLS-ER | 45.76 | 29.33 | 16.19 | 33.08 | 30.74 | 67.10 | 46.44 | 30.26 | 80.29 | 97.62 | 47.68 |
| RanPAC | 92.64 | 70.87 | 43.75 | 56.13 | 51.83 | 88.08 | 83.51 | 40.34 | 86.74 | 96.88 | 71.08 |
| MoE | 43.01 | 0.94 | 55.48 | 25.16 | 74.22 | 97.67 | 84.57 | 33.38 | **96.88** | **99.90** | 61.12 |
| L2P | 10.88 | 1.45 | 0.97 | 4.04 | 14.55 | 12.17 | 22.48 | 9.16 | 89.69 | 98.62 | 26.40 |
| DAP | 9.93 | 1.07 | 1.84 | 17.65 | 15.74 | 15.44 | 22.19 | 16.34 | 86.20 | 99.44 | 28.48 |
| D-Prompt | 10.61 | 1.31 | 1.44 | 3.09 | 17.03 | 23.83 | 25.03 | 14.77 | 93.42 | 99.33 | 28.99 |
| C-Prompt | 11.66 | 0.67 | 0.62 | 0.33 | 1.72 | 13.04 | 6.14 | 16.62 | 21.74 | 96.21 | 16.88 |
| Ours | **95.89** | **81.25** | **69.57** | **84.12** | **85.30** | **98.56** | **92.92** | **45.45** | 96.60 | 99.30 | **84.90** |

over previous feature distributions through Eq. (12). As a result, the selected expert is employed to initialize a new expert.

**Step 3: Interactive optimization with alignment constraints.** Once the new expert $\mathcal{E}_j$ is created, we perform joint optimization incorporating mutual information-based prediction alignment via Eq. (5) and KL-divergence feature alignment using Eq. (7). Furthermore, we calculate the HSIC regularization losses to encourage disentanglement between global and local backbones using Eq. (9). All regularization terms are incorporated into the main objective function for expert optimization :

$$\mathcal{L}_{\text{final}}(\mathbf{X}) = -\sum_{c=1}^{b}\sum_{t=1}^{C}\left\{\mathbf{y}_c[t]\log(F_c(\mathcal{E}_j, \mathbf{x}_c)[t])\right\} + \lambda_1\mathcal{L}_{\text{MI}} \\ + \lambda_2\mathcal{L}_{\text{KLDBFA}} + \lambda_3\mathcal{L}_{HSIC}(Q, B, \mathbb{P}_{\mathbf{Z}^g, \mathbf{Z}^l}), \tag{13}$$

where $F_c(\mathcal{E}_j, \mathbf{x}_c)[t]$ denotes the predicted probability of class $t$ for sample $\mathbf{x}_c$. $\lambda_1$, $\lambda_2$, $\lambda_3$ are trade-off hyperparameters balancing different loss components. The model parameters $\{\theta^g, \theta^l, \varphi_j^f, \varphi_j^c\}$ is updated using Eq. (13). The detailed algorithm is summarized in **Appendix-B** from SM.

# 4  Experiment

## 4.1  Experimental Setup

**Metrics.** In the context of the MDCL scenarios, we assess and compare model efficacy at the final task through two key metrics: the classification accuracy of a single domain (e.g., **C10** or **CUB200**) and the overall performance across all domains (**Avg**).

**Datasets.** The datasets used in our experiment can be logically categorized into three primary fields according to [21]. **Natural Domains** include CIFAR-10 [25] (C10), TinyImageNet [26] (TImg), CUB-200 [39], MNIST [27] and ImageNet-R [17] (ImgR), covering a range of tasks from basic image classification to fine-grained recognition and robustness testing across various visual styles. **Aerial Domains** comprise EuroSAT [16] and RESISC45 [7], focusing on satellite imagery for land cover classification and environmental monitoring. **Medical Domains** consist of CropDiseases [31] (Disease) and ChestX [43], specialized for identifying plant diseases and diagnosing medical conditions through radiographic images, respectively. Then, we randomly shuffle these datasets to construct three highly challenging MDCL scenarios (**CDM**, **ETI** and **TRC** which are derived from the initial letters of the first three domains). Detailed experimental configurations are provided in **Appendix-C** from SM.

Table 3: Comparison of Baselines in terms of parameter and computational efficiency (**in CDM**).

| Methods | Train Params↓ | Iter/s↑ | GPU Avg↓ | GPU Max↓ | CPU Avg↓ | CPU Max↓ |
|---|---|---|---|---|---|---|
| DER++(Re) | 42.84M | 1.04 | 12919.37MB | 12919.37MB | 9756.08MB | 16805.70MB |
| CLS-ER | 42.84M | 2.13 | 6199.39MB | 6199.39MB | 9658.21MB | 16700.56MB |
| RanPAC | 1.19M | 4.27 | 3065.27MB | 3087.97MB | 9823.56MB | 17465.18MB |
| MoE | 4.03M | 1.77 | 11098.88MB | 11098.88MB | 14203.61MB | 17262.77MB |
| L2P | **0.20M** | **5.08** | 3420.06MB | 3420.06MB | 9654.75MB | 16798.61MB |
| DAP | 0.51M | 2.76 | 3686.95MB | 3686.95MB | 9911.54MB | 16958.32MB |
| D-Prompt | 0.41M | 3.11 | 3556.64MB | 3556.64MB | 9866.40MB | 17040.50MB |
| C-Prompt | 3.99M | 4.91 | 4775.65MB | 4775.65MB | 9869.61MB | 16941.20MB |
| Ours | 42.54M | 3.06 | **2926.20MB** | **2926.20MB** | **9525.63MB** | **16681.54MB** |

Table 4: Impact of individual and combined components on model performance in ETI.

| Methods | EuroSAT | TImg | ImgR | CUB200 | C100 | MNIST | RESISC45 | ChestX | C10 | Disease | Avg |
|---|---|---|---|---|---|---|---|---|---|---|---|
| CB | 19.59 | 8.37 | 11.81 | 12.29 | 36.92 | 49.88 | 61.14 | 30.04 | 92.79 | 99.00 | 42.18 |
| SBE | 85.38 | 70.49 | 55.42 | 73.45 | 77.05 | 90.05 | 85.93 | 37.73 | 93.65 | 99.13 | 76.83 |
| CBE | 92.38 | 76.49 | 62.42 | 78.45 | 82.05 | 95.05 | 90.93 | 38.73 | 94.17 | 99.04 | 80.97 |
| CBE+MI | 95.08 | 80.93 | 68.19 | 82.49 | 84.68 | 97.15 | 91.29 | 39.56 | 96.18 | 98.28 | 83.38 |
| CBE+KL | 92.18 | 78.02 | 66.13 | 78.97 | 82.50 | 92.24 | 90.54 | 44.58 | 95.81 | 99.13 | 82.01 |
| CBE+HSIC | 93.65 | 76.97 | 64.37 | 78.56 | 82.35 | 91.88 | 91.29 | 45.53 | 95.49 | 99.06 | 81.92 |
| CBE+MI+KL | 95.35 | **81.61** | 68.30 | 83.56 | **85.54** | 97.43 | 92.28 | **46.44** | 96.22 | 99.01 | 84.57 |
| CBE+MI+HSIC | 95.87 | 80.85 | 68.74 | 83.15 | 85.52 | 98.03 | 92.08 | 42.43 | 96.33 | 98.91 | 84.19 |
| CBE+KL+HSIC | 93.24 | 78.22 | 66.45 | 80.49 | 82.55 | 92.21 | 91.39 | 45.26 | 95.59 | 99.08 | 82.45 |
| LEAR | **95.89** | 81.25 | **69.57** | **84.12** | 85.30 | **98.56** | **92.92** | 45.45 | **96.60** | **99.30** | **84.90** |
| LEAR w/o ESM | 3.72 | 1.25 | 2.36 | 83.95 | 8.24 | 4.91 | 14.63 | 1.05 | 17.85 | 99.15 | 23.71 |

## 4.2 Experimental Results

**Results in Multi-Domain Continual Learning.** Our comprehensive evaluation compares LEAR against state-of-the-art approaches across three distinct domain sequences (Tables 1 and 2). As shown in Table 1, LEAR achieves an outstanding average accuracy of 84.61% in the CDM scenario. Specifically, LEAR outperforms the replay-based DER++ (Refresh) by 55.39%, highlighting its superior ability to mitigate catastrophic forgetting without requiring memory buffers. When compared to expansion-based methods, LEAR shows substantial advantages over both RanPAC (71.01%) and MoE-adapters (57.33%), particularly in challenging domains like TinyImageNet and ChestX, while maintaining consistent performance across all evaluated domains.

The reshuffled domain sequence in Table 2 further validates LEAR's adaptability, where it achieves an even higher average accuracy of 84.90% in the ETI scenario, outperforms dual-branch method CLS-ER (47.68%) by 37.22%, with especially large gap on ImageNet-R. Moreover, LEAR demonstrates over 55% higher average accuracy than the domain-incremental method DAP (28.48%) and other listed prompt-based approaches. These results highlight LEAR's ability to effectively learn and retain knowledge regardless of the domain order.

Moreover, as shown in Figure 2 (a), the proposed LEAR achieves the lowest forgetting rate in CDM scenario compared to alternative methods. As the number of tasks increases, LEAR consistently maintains stable and superior performance across domains with various fields and different complexities, effectively addressing the catastrophic forgetting prevalent in existing approaches. Detailed results for the TRC scenario are provided in the **Appendix-D** from SM.

## 4.3 Ablation Studies

**The impact of components in LEAR.** Table 4 provides empirical validation for the theoretical contributions of each proposed module. Where "CB" denotes using only the collaborative backbone with a single shared expert network across all data domains; "CBE" extends CB with task-specific expert network expansion and ESM expert selection; "SBE" denotes the configuration where "CBE"'s dual backbone architecture is replaced with a single backbone; and "CBE+MI/KL/HSIC" represents CBE augmented with individual components or combination of components. "LEAR w/o ESM" denotes randomly selecting experts during the beginning and testing phase of each task. This table

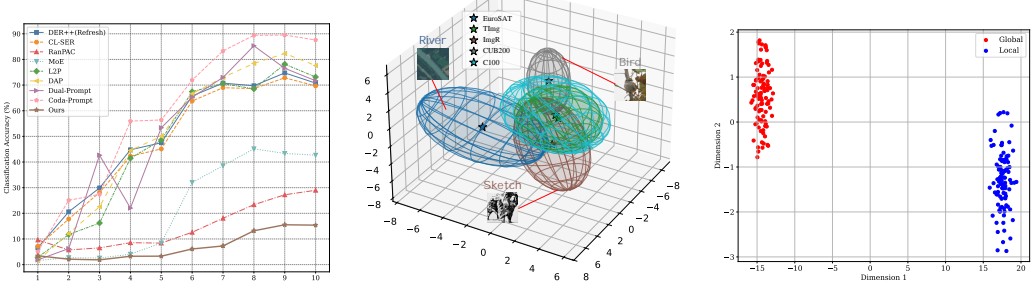

(a) The forgetting curve of CDM.    (b) Visualization of distributions.    (c) Clustering results of features.

Figure 2: (a): The comparison of the forgetting curves between LEAR and other baseline methods after learning the CDM sequence. (b): Visualization of memory distributions from the ETI sequence after PCA dimensionality reduction, showing mean positions and covariance ellipsoids. (c): Feature visualization of global backbone and local backbone with HSICBCO regularization.

demonstrates that the collaborative backbone design ("SBE"->"CBE") and the dynamically expanded expert network ("CB"->"CBE") significantly enhance model performance in ETI. In addition, each regularization term and its combinations contribute to varying degrees of performance improvement over "CBE" on these two sequences. Furthermore, the model's performance will significantly drop when inappropriate experts are chosen ("LEAR"->"LEAR w/o ESM"), thereby demonstrating the necessity of ESM. These evaluation results align well with our methodological design.

**The analysis of parameter and computational efficiency.** As shown in Table 3, LEAR achieves the lowest GPU and CPU utilization among all baseline methods. While baselines including prompt-based approaches (e.g., DualPrompt) and adapter-based variants (e.g., MoE-Adapters) indeed contain fewer trainable parameters (0.5%-5% per ViT block), they inevitably require complete backpropagation through all ViT blocks, necessitating the storage of intermediate activations throughout the entire backbone due to chain rule dependencies, which maintain extensive computation graphs. Conversely, LEAR's innovative design strategically fine-tunes only the last three ViT layers and terminates backpropagation after the third-last layer, thereby significantly reducing the computation graph and GPU usage.

**Visualization of the Expert Selection Mechanism.** Figure 2b shows the first five ESM-preserved distributions from the ETI scenario, visualized in 3D space after PCA reduction. ESM computes Mahalanobis distances between these distributions and incoming task samples to select experts for either network expansion or test evaluation.

**Visualization of the HSICBCO approach.** Both the global and local backbones are initialized with identical pretrained weights. Under the guidance of MIBPA and KLDBFA, they learn task-general and task-specific representations, respectively. However, their feature representations still exhibit strong correlations. As illustrated in Figure 2c, the proposed HSICBCO module effectively decouples these representations, demonstrating its capability to promote distinct feature learning. Additional ablation results are provided in the **Appendix-D** from SM.

## 5    Conclusion and Limitation

In this paper, we propose LEAR, a novel framework for Multi-Domain Continual Learning that simultaneously addresses stability plasticity and efficiency. Specifically, built on a collaborative backbone structure, we introduce MIBPA and KLDBFA to maintain historical prediction consistency and task-specific feature alignment during model updates, while HSICBCO ensures disentangled and complementary representations. Additionally, ESM dynamically selects relevant experts for efficient network expansion and task-agnostic prediction. The empirical results demonstrate the effectiveness of the proposed approach. The primary limitation of this paper is that the proposed approach would contain a considerable number of parameters. To address this issue, we will propose a novel expert merging technology with self-distillation for effective model compression.

## Acknowledgements

This work was supported by the Sichuan Provincial Natural Science Foundation Project (No.2025ZNSFSC0510) and National Natural Science Foundation of China (Grant No: 62506067)

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
