# OpenReview forum: "Learning Expandable and Adaptable Representations for Continual Learning"
_NeurIPS.cc/2025/Conference — NeurIPS 2025 poster_

### Official Review · Reviewer_tTuR · 2025-06-22

**Clarity:** 3
**Significance:** 2
**Originality:** 2
**Rating:** 4
**Confidence:** 4

**Summary:**

This work focuses on addressing catastrophic forgetting on evolving data domains of continual learning, which is multi-domain continual learning. This work introduces dynamic expansion approach comprising global and local backbones. The global backbone is trained using a mutual information metric. To mitigate forgetting, HSIC-based method is used. Also, this framework includes expert selection to identify the most relevant expert according to the input distribution.

**Questions:**

1. Which pre-trained ViT is used in the proposed framework, does the comparing methods using the same pre-trained model?

2. In Table 3, the number of training parameters of the proposed framework is among the largest, but how does the GPU utilization is the least? The section that refers to Table 3 is only reiterating the superiority of the framework. I don’t see an explanation of this in the text.

3. Also, how would it be that the proposed framework has the almost comparable number of training parameters with DER++ and CLS-ER? From my understanding, the proposed framework only trains the lightweight experts and the last layers of the backbones.

4. On line 175, what is the “concentration operation”? Isn’t it “concatenation”? If it is concatenation, the symbol should be $\bigoplus$.

**Ethical Concerns:**

["NO or VERY MINOR ethics concerns only"]

**Final Justification:**

The authors' rebuttal addresses most of my concerns. However, it seems irresponsible to omit a key part of the code implementation in the supplementary material when the code is provided, even though the authors have clarified the implementation in the rebuttal.
I have raised my rating accordingly.

**Limitations:**

See weaknesses.

**Quality:**

2

**Strengths And Weaknesses:**

Strengths

1.	The writing is clear.
2.	The motivation of the global and local backbones seems acceptable.

Weaknesses

1.	In Section 3.4, the KL divergence is applied between two feature distributions. I don’t think it is a good idea to use KL divergence as a metric between two feature distributions, because the KL divergence is usually used on predictions (or logits, probabilities). Features are in high dimensions, I don’t know how you get the distributions of the features exactly (Actually I went through the code in the supplementary material, and did not find the specific implementation), but if you just take the softmax on the features to get the probabilities, most of the values would be close to zero. This provides little information. I think this section needs more elaboration.
2.	The results in Table 1 shows that the framework achieve better performance than other methods by a large margin. But as I see the results on the last three rows, I wonder if there are results of another baseline method that does not use any of the three alignments. Because it seems the three alignments are not too significant in terms of achieving the competitive results. This leads to another concern that is the main contribution of the improvement comes from the extra pre-trained backbone, the larger pre-trained backbone or the alignments?

---

> ### Author Rebuttal · Authors · 2025-07-31
>
> We sincerely appreciate your positive feedback regarding the clarity of our writing and the validity of our collaborative backbone design motivation.
>
> For weaknesses and questions, we provide our feedback as follows.
>
> **Q1: Implement of KLDBFA.**
>
> **A1:** We clarify that we approximate the feature distributions from both the Local backbone and Frozen Local backbone as multivariate Gaussian distributions, for which we derive closed-form solutions of their KL divergence. Actually, many modern generative model evaluation metrics (FID, Kernel MMD) assume that real and generated data can be approximated by Gaussian distributions in a high-dimensional feature space. Therefore, we can model the feature distributions from both backbones for a data batch as Gaussian distributions.  The rationale for choosing KL divergence over alternative distributional metrics is because: (1) Unlike symmetric metrics such as Jensen-Shannon divergence, KL divergence's directional nature allows us to specifically constrain the current feature distribution to remain close to the historical distribution; (2) The computational overhead of distances like MMD and Wasserstein distance is relatively high.
>
> In code implementation, we separately compute the mean vectors and covariance matrices of the feature vectors output by the Local Backbone and the Frozen Local Backbone, and then calculate the KL divergence between these two distributions $P({\bf Z}^l)=\mathcal{N}(\boldsymbol{\mu}_1, \boldsymbol{\Sigma}_1)$ and $P({{\hat{\bf Z}}}^l)=\mathcal{N}(\boldsymbol{\mu}_2, \boldsymbol{\Sigma}_2)$ ($\boldsymbol{\mu}$ and $\boldsymbol{\Sigma}$ denotes mean vectors and covariance matrices, respectively) using the following equation:
>
> $$D_{\mathrm{KL}}(P({\bf Z}^l) \,||\,  P({\hat{\bf Z}}^l) ) = \frac{1}{2} \left[\log \left( \frac{\det \boldsymbol{\Sigma}_2}{\det \boldsymbol{\Sigma}_1} \right)- d + \mathrm{tr}(\boldsymbol{\Sigma}_2^{-1} \boldsymbol{\Sigma}_1)+ (\boldsymbol{\mu}_2 - \boldsymbol{\mu}_1)^\top \boldsymbol{\Sigma}_2^{-1} (\boldsymbol{\mu}_2 - \boldsymbol{\mu}_1)\right]$$
>
> where 'det', 'd', and 'tr' denote the determinant, dimension, and trace of a matrix, respectively.
>
> **Q2: The source of LEAR performance improvement and collaborative backbone design independent performance.**
>
> **A2:**  We clarify that we did not employ larger pre-trained backbones compared to baseline methods to enhance performance. Specifically, the performance improvement stems from three synergistic components: (1) the collaborative global-local backbone structure, (2) the task-specific dynamically expandable expert networks, and (3) the proposed three regularizations terms.
>
> We note that all four reviewers raise concerns in the individual contributions of LEAR's components and have provided consistent suggestions for improving our ablation studies. Recognizing the importance of this issue, we have conducted extensive additional experiments on ETI and TRC sequences during the rebuttal phase and reorganized the ablation study tables accordingly:
>
> ### Table 1: Results of revised ablation studies
> |Methods|EuroSAT|TImg|ImgR|CUB200|C100|MNIST|RESISC45|ChestX|C10|Disease|Avg|
> |-|-|-|-|-|-|-|-|-|-|-|-|
> |CB|19.59|8.37|11.81|12.29|36.92|49.88|61.14|30.04|92.79|99.00|42.18|
> |SBE|85.38|70.49|55.42|73.45|77.05|90.05|85.93|37.73|93.65|99.13|76.83
> |CBE|92.38|76.49|62.42|78.45|82.05|95.05|90.93|38.73|94.17|99.04|80.97|
> |CBE+MI|95.08|80.93|68.19|82.49|84.68|97.15|91.29|39.56|96.18|98.28|83.38|
> |CBE+KL|92.18|78.02|66.13|78.97|82.50|92.24|90.54|44.58|95.81|99.13|82.01|
> |CBE+HSIC|93.65|76.97|64.37|78.56|82.35|91.88|91.29|45.53|95.49|99.06|81.92|
> |CBE+MI+KL|95.35|81.61|68.30|83.56|85.54|97.43|92.28|46.44|96.22|99.01|84.57|
> |CBE+MI+HSIC|95.87|80.85|68.74|83.15|85.52|98.03|92.08|42.43|96.33|98.91|84.19|
> |CBE+KL+HSIC|93.24|78.22|66.45|80.49|82.55|92.21|91.39|45.26|95.59|99.08|82.45|
> |LEAR|95.89|81.25|69.57|84.12|85.30|98.56|92.92|45.45|96.60|99.30|84.90|
>
>
> |Methods|TImg|RESISC45|CUB200|ChestX|ImgR|EuroSAT|MNIST|C10|Disease|C100|Avg|
> |-|-|-|-|-|-|-|-|-|-|-|-|
> |CB|8.43|22.67|39.45|23.51|25.67|42.74|60.95|64.50|93.45|84.87|46.62|
> |SBE|72.58|81.94|70.12|32.93|57.85|91.19|89.80|89.66|93.96|84.59|76.46
> |CBE|79.01|89.72|80.68|30.26|64.06|92.34|94.15|94.44|97.05|86.19|80.79|
> |CBE+MI|81.24|90.76|82.51|40.13|67.98|95.19|97.68|95.64|98.03|85.77|83.49|
> |CBE+KL|78.77|90.36|80.22|40.21|66.52|94.67|97.01|95.27|98.40|87.02|82.85|
> |CBE+HSIC|79.14|90.21|80.54|42.12|66.21|93.26|94.24|94.48|97.25|85.41|82.29|
> |CBE+MI+KL|81.24|90.39|82.60|44.11|68.58|96.01|98.37|96.22|98.86|86.17|84.25|
> |CBE+MI+HSIC|82.21|90.43|83.05|44.74|68.41|95.18|95.86|96.09|98.91|85.85|84.07|
> |CBE+KL+HSIC|79.67|89.88|81.44|42.47|66.27|95.07|96.23|95.54|98.91|86.74|83.22|
> |LEAR|80.30|92.41|83.49|44.67|69.28|96.62|98.53|95.84|99.16|86.22|84.65|
>
> Where ''CB'' denotes using only the collaborative backbone with a single shared expert network across all datasets; ''CBE'' extends CB with task-specific expert network expansion and ESM expert selection; ''SBE'' denotes the configuration where ''CBE'''s dual backbone architecture is replaced with a single backbone; and "CBE+MI/KL/HSIC" represents CBE augmented with individual components or combination of components.
>
> This table demonstrates that the collaborative backbone design (''SBE''->''CBE'') and the dynamically expanded expert network (''CB''->''CBE'') significantly enhances model performance on both ETI and TRC. In addition, each regularization term and their combinations contribute to varying degrees of performance improvement over ''CBE'' on these two sequences.
>
> **Q3: The use of pre-trained ViT.**
>
> **A3:** We initialize both the global backbone and the local backbone using a ViT-B/16 model pre-trained on ImageNet-21K. All comparison methods used the same pre-trained model, except for the MoEAdapter which adopted the CLIP ViT-B/16.
>
> **Q4: The relation between the number training parameters and the GPU usage.**
>
> **A4:** We would like to clarify that GPU memory consumption in our experiment setting is not solely determined by the number of trainable parameters; it can stem from multiple sources, including the size of the pre-trained model used, additional components such as adapters or prompts introduced for continual learning, and the use of memory buffers. For instance, although MoE Adapter has a relatively small number of trainable parameters, it employs the CLIP model as its backbone. Furthermore, the majority of LEAR's trainable parameters originate from fine-tuned layers in the ViT backbone, resulting in substantial GPU usage overlap with the original frozen ViT. Additionally, gradient information is not preserved during training. Therefore, the low GPU memory usage of LEAR is both reasonable and expected.
>
> **Q5: Why LEAR has the almost comparable number of training parameters with DER++ and CLS-ER.**
>
> **A5:** We clarify that both DER++(Rresh) [1] and CLS-ER [2] were originally implemented using a single ResNet-18 backbone without frozen parameters. To ensure fair comparison when migrating these baselines to ViT, we adopt the consistent backbone configurations between LEAR (keep last three layers of both backbones trainable) and other baselines (DER++(Rresh) and CLS-ER). This results in comparable scales of trainable parameters across all three methods.
>
> In practice, we have also conducted comparative experiments by activating the last three layers of ViT backbones for other baselines, yielding the following results:
>
> |Methods|EuroSAT|TImg|ImgR|CUB200|C100|MNIST|RESISC45|ChestX|C10|Disease|Avg|
> |-|-|-|-|-|-|-|-|-|-|-|-|
> |RanPAC+FT3|97.83|44.49|21.44|16.76|22.05|87.94|78.94|40.06|74.41|95.84|57.97|
> |L2P+FT3|11.15|2.06|1.50|1.90|14.97|21.12|31.94|11.72|89.01|98.32|28.37|
> |DAP+FT3|14.86|0.83|2.02|4.66|18.48|30.66|27.34|15.91|90.01|98.91|30.37|
> |DualP+FT3|12.69|0.69|0.30|0.69|0.93|14.53|3.09|15.84|25.55|96.72|17.10|
> |CODAP+FT3|14.16|0.51|1.02|0.64|2.30|13.02|8.80|9.52|39.79|95.61|18.54|
> |LEAR+FT3|95.89|81.25|69.57|84.12|85.30|98.56|92.92|45.45|96.60|99.30|84.90|
>
> where ''FTX'' denotes that we activate the final ''X'' layers of the ViT backbone. These results indicate that merely scaling up model parameters cannot guarantee performance gains in MDCL scenario, which necessitates the incorporation of regularization constraints during backbone adaptation to mitigate adverse parameter optimization caused by domain shift.
>
> In addition, we further considered how to reduce the number of training parameters. We tried fine-tuning only the MHSA and MLP sublayers in the last three layers of the VIT backbone:
>
> |Methods|TImg|RESISC45|CUB200|ChestX|ImgR|EuroSAT|MNIST|C10|Disease|C100|Avg|TrainParms
> |-|-|-|-|-|-|-|-|-|-|-|-|-|
> |FT1|80.42|90.02|82.68|37.14|65.76|95.21|97.17|95.62|98.54|84.17|82.67|14.26M
> |FT2|81.06|90.54|82.53|38.92|67.63|95.37|98.17|95.94|98.90|85.41|83.45|28.43M
> |FT3|80.30|92.41|83.49|44.67|69.28|96.62|98.53|95.84|99.16|86.22|84.65|42.54M
> |FT3MHSA|81.04|92.36|83.22|44.18|69.03|96.95|98.45|96.13|99.18|85.70|84.62|14.27M
> |FT3MLP|80.75|90.94|82.68|42.40|68.97|96.32|97.53|95.88|98.91|85.65|84.00|28.43M
> |FT4|81.65|91.76|82.87|44.89|69.94|96.18|97.63|96.43|99.14|86.80|84.73|56.80M
>
> These results demonstrate that fine-tuning only the MHSA achieves comparable or even better performance than fine-tuning the entire block. This move reduced the total number of parameters to 14.27M, a significant reduction of 28.27M compared to the original.
>
> **Q6: Typo: concentration operation.**
>
> **A6:** Thank you for highlighting this oversight. It is indeed a concatenation operation, we have corrected the word and symbol in the revised paper.
>
> ### References:
>
> [1]A unified and general framework for continual learning//ICLR24
>
> [2]Learning fast, learning slow: A general continual learning method based on complementary learning system//ICLR22

---

> ### Author Response · Authors · 2025-08-08
> **Supplemental Explanations for Reviewer tTuR's Comments**
>
> Dear Reviewer tTuR,
>
> We sincerely appreciate that you may be occupied with other commitments and thus unable to provide a response to our rebuttal at this time. In light of this, we provide further explanations for your previous comments. We hope these supplemental analyses meet with your approval.
>
> **Regarding KLDBFA**, While KL divergence traditionally applied to prediction probabilities, it can effectively regularize feature distributions under a parametric distributional assumption, such as Gaussian distributions, as implemented in Variational Autoencoders (VAEs). In the implement of KLDBFA, We derive closed-form solutions for the KL divergence between the estimated feature distributions of the frozen local backbone and those of the local backbone to align historical and current representation distributions at the feature level. In addition, the t-SNE visualization results presented in Figure 1 of the Appendix from Supplementary Materials demonstrate unimodal patterns in the feature representations extracted from two MDCL sequences, thereby providing additional support for modeling these representations as Gaussian distributions.
>
> **Concerning analyses of performance gain**, following all four reviewers' collective suggestions, we have reorganized the ablation studies to systematically evaluate component contributions in LEAR. The results have demonstrated that our proposed:
>
> (1) collaborative backbone architecture,
>
> (2) task-specific expert networks,
>
> (3) three regularization terms (MIBPA, KLDBFA, HSICBCO), and
>
> (4) Expert Selection Mechanism (ESM)
>
> collectively ensure LEAR’s superior performance in MDCL scenarios, achieving optimal balance between plasticity, stability and efficiency.
>
> **Regarding parameter count and GPU usage**, We provide a more detailed technical elaboration:
>
> While baseline methods including prompt-based approaches (e.g., DualPrompt) and adapter-based methods (e.g., MoE-Adapters) indeed contain fewer trainable parameters (0.5%-5% of ViT-B/16 backbone), they inevitably require complete backpropagation through all ViT layers, necessitating the storage of intermediate activations throughout the entire vit backbone due to chain rule dependencies, which maintain extensive computation graphs;
>
> conversely, LEAR's innovative design strategically fine-tunes only the last three ViT layers while terminating backpropagation after the third-last layer, thereby significantly reducing computation graph and GPU usage.
>
> We sincerely look forward to receiving any additional suggestions you may have. We are fully committed to addressing all of your concerns, which will undoubtedly enhance the quality and impact of our work.

---

> ### Author Response · Authors · 2025-08-08
> **Reorganized Ablation Studies for Reviewer tTuR**
>
> We are grateful for your valuable feedback on our work. We have reorganized the ablation studies below to **systematically evaluate component contributions**:
>
> ### Table 1: Effectiveness of different component combinations in LEAR
> |Methods|EuroSAT|TImg|ImgR|CUB200|C100|MNIST|RESISC45|ChestX|C10|Disease|Avg|
> |-|-|-|-|-|-|-|-|-|-|-|-|
> |UpperBound|96.71|82.77|72.81|85.08|86.39|98.97|93.72|50.14|96.49|99.35|86.24
> |CB|19.59|08.37|11.81|12.29|36.92|49.88|61.14|30.04|92.79|99.00|42.18
> |SBE|85.38|70.49|55.42|73.45|77.05|90.05|85.93|37.73|93.65|99.13|76.83
> |CBE|92.38|76.49|62.42|78.45|82.05|95.05|90.93|38.73|94.17|99.04|80.97
> |CBE+MI|95.08|80.93|68.19|82.49|84.68|97.15|91.29|39.56|96.18|98.28|83.38
> |CBE+KL|92.18|78.02|66.13|78.97|82.50|92.24|90.54|44.58|95.81|99.13|82.01
> |CBE+HSIC|93.65|76.97|64.37|78.56|82.35|91.88|91.29|45.53|95.49|99.06|81.92
> |CBE+MI+KL|95.35|81.61|68.30|83.56|85.54|97.43|92.28|46.44|96.22|99.01|84.57
> |CBE+MI+HSIC|95.87|80.85|68.74|83.15|85.52|98.03|92.08|42.43|96.33|98.91|84.19
> |CBE+KL+HSIC|93.24|78.22|66.45|80.49|82.55|92.21|91.39|45.26|95.59|99.08|82.45
> |LEAR|95.89|81.25|69.57|84.12|85.30|98.56|92.92|45.45|96.60|99.30|84.90
> |LEAR w/o ESM|03.72|01.25|02.36|83.95|08.24|04.91|14.63|01.05|17.85|99.15|23.71
>
>
> |Methods|TImg|RESISC45|CUB200|ChestX|ImgR|EuroSAT|MNIST|C10|Disease|C100|Avg|
> |-|-|-|-|-|-|-|-|-|-|-|-|
> |UpperBound|82.77|93.72|85.08|50.14|72.81|96.71|98.97|96.49|99.35|86.39|86.24
> |CB|08.43|22.67|39.45|23.51|25.67|42.74|60.95|64.50|93.45|84.87|46.62
> |SBE|72.58|81.94|70.12|32.93|57.85|91.19|89.80|89.66|93.96|84.59|76.46
> |CBE|79.01|89.72|80.68|30.26|64.06|92.34|94.15|94.44|97.05|86.19|80.79
> |CBE+MI|81.24|90.76|82.51|40.13|67.98|95.19|97.68|95.64|98.03|85.77|83.49
> |CBE+KL|78.77|90.36|80.22|40.21|66.52|94.67|97.01|95.27|98.40|87.02|82.85
> |CBE+HSIC|79.14|90.21|80.54|42.12|66.21|93.26|94.24|94.48|97.25|85.41|82.29
> |CBE+MI+KL|81.24|90.39|82.60|44.11|68.58|96.01|98.37|96.22|98.86|86.17|84.25
> |CBE+MI+HSIC|82.21|90.43|83.05|44.74|68.41|95.18|95.86|96.09|98.91|85.85|84.07
> |CBE+KL+HSIC|79.67|89.88|81.44|42.47|66.27|95.07|96.23|95.54|98.91|86.74|83.22
> |LEAR|80.30|92.41|83.49|44.67|69.28|96.62|98.53|95.84|99.16|86.22|84.65
> |LEAR w/o ESM|04.27|04.83|68.49|09.42|03.76|14.59|01.98|06.33|01.64|09.25|12.46
>
> Table 1 clearly demonstrates the contribution of each module in LEAR:
>
> (1) ''UpperBound'' denotes the results in a single-domain setting, which closely approximates LEAR's performance upper bound in MDCL;
>
> (2) ''CB'' denotes using only the collaborative backbone with a single shared expert network across all datasets;
>
> (3) ''CBE'' extends ''CB'' with task-specific expert network expansion and ESM expert selection. The results demonstrate that the dynamic expert network expansion and selection (''CB''->''CBE'') significantly enhance model performance on both ETI and TRC;
>
> (4) ''SBE'' denotes the configuration where ''CBE'''s dual-backbone architecture is replaced with a single backbone. The dual-backbone design yields about a 4% performance gain over the single-backbone counterpart (''SBE''->''CBE'');
>
> (5) "CBE+MI/KL/HSIC" represents ''CBE'' augmented with individual components or combination of components. Each regularization component and its respective combinations yield effective performance enhancements relative to ''CBE'' on both sequences;
>
> (6) "LEAR" denotes the complete framework, while "LEAR w/o ESM" represents the variant where experts are randomly selected during both the initialization and testing phases of each task. This configuration explains the observed significant performance degradation ("LEAR"->"LEAR w/o ESM")
>
> ### Table 2: Performance comparison of different backbone fine-tuning configurations in LEAR​
> |Methods|EuroSAT|TImg|ImgR|CUB200|C100|MNIST|RESISC45|ChestX|C10|Disease|Avg|TrainParms
> |-|-|-|-|-|-|-|-|-|-|-|-|-|
> |FT1|94.52|79.83|65.42|80.92|82.89|97.29|90.51|40.90|95.37|98.29|82.60|14.26M
> |FT2|94.90|80.94|67.05|83.21|84.67|98.16|90.63|44.31|95.78|98.82|83.85|28.43M
> |FT3|95.89|81.25|69.57|84.12|85.30|98.56|92.92|45.45|96.60|99.30|84.90|42.54M
> |FT4|96.01|81.96|69.44|83.80|86.05|98.49|92.98|47.86|96.34|99.22|85.22|56.80M
>
> |Methods|TImg|RESISC45|CUB200|ChestX|ImgR|EuroSAT|MNIST|C10|Disease|C100|Avg|TrainParms
> |-|-|-|-|-|-|-|-|-|-|-|-|-|
> |FT1|80.42|90.02|82.68|37.14|65.76|95.21|97.17|95.62|98.54|84.17|82.67|14.26M
> |FT2|81.06|90.54|82.53|38.92|67.63|95.37|98.17|95.94|98.90|85.41|83.45|28.43M
> |FT3|80.30|92.41|83.49|44.67|69.28|96.62|98.53|95.84|99.16|86.22|84.65|42.54M
> |FT4|81.65|91.76|82.87|44.89|69.94|96.18|97.63|96.43|99.14|86.80|84.73|56.80M
>
> We conducted comparative experiments on different fine-tuning configurations for the collaborative backbones of LEAR, where "FTX" denotes fine-tuning the last X layers while freezing preceding backbone layers. As shown in Table 2, the configuration utilizing the last three trainable layers ("FT3") of the collaborative backbones achieves robust performance with limited parameter growth compared to alternative configurations.

---

### Official Review · Reviewer_NZTA · 2025-06-28

**Clarity:** 3
**Significance:** 3
**Originality:** 2
**Rating:** 4
**Confidence:** 4

**Summary:**

This paper proposes LEAR to address multi-domain continual learning that jointly addresses the stability–plasticity trade-off. The approach incorporates a collaborative backbone along with three key components: MIBPA and KLDBFA for maintaining historical knowledge consistency and task-specific feature alignment, and HSICBCO for learning disentangled representations. An expert selection module (ESM) dynamically activates relevant experts to enable scalable network expansion and efficient inference. Experimental results validate the effectiveness of the proposed method.

**Questions:**

Please refer to the weaknesses.

**Ethical Concerns:**

["NO or VERY MINOR ethics concerns only"]

**Final Justification:**

I will maintain my original score. I believe that this work requires substantial revisions before it is ready for publication. In particular, the paper should place more emphasis on the truly novel components and reduce the focus on the more straightforward applications of existing techniques.

**Limitations:**

The authors have addressed the limitations and potential negative societal impact of their work in Section 5.

**Quality:**

2

**Strengths And Weaknesses:**

### Strengths:
+ The paper addresses a new and relatively unexplored problem setup.
+ The proposed method appears to outperform prior methods significantly under the new setting.
+ The paper is well-structured and includes a detailed empirical study.

### Weaknesses:
+ Despite the claim of proposing several new techniques, many of the components seem to be straightforward applications of existing techniques in continual learning. For instance:
   + The use of a dual-branch architecture is a well-established strategy in the continual learning literature.
   + The proposed LEAR module, which learns task-specific experts or adaptors, closely resembles techniques used to build expandable representations across tasks.
   + The idea behind MIBPA (maintaining consistency with historical predictions) is a well-known strategy for mitigating forgetting in continual learning. In this work, the authors adopt mutual information as the consistency metric instead of the more common cross-entropy. However, it's unclear whether this choice offers any real advantage. Since mutual information is symmetric and generally less sensitive to low-confidence outputs than cross-entropy, it's not obvious why it would be preferable in this context. A theoretical or empirical comparison between the two would help clarify whether this design decision is justified.
+ Based on the discussion above, I encourage the authors to clearly delineate which components are truly novel and which are adaptations of known techniques.

+ Equation (7) treats the outputs of the two branches as random variables, but the formulation lacks detail. What assumptions are made about their distributions? Is there a specific probabilistic model behind this formulation?
+ The task definition needs further clarification. Does each task correspond to a specific domain or dataset? Or, is the number of classes per task fixed? If the former, I would also be interested in how the method performs under the latter setup, which arguably reflects a more practical scenario.
+ The ablation in Table 1 could be more informative. Instead of removing components in parallel, it would be better to add or remove them step by step. This would make it clearer how each module contributes to the final performance. The current setup doesn’t fully show the relative importance of each part.
+ Although the authors append code in the supplementary materials, the provided code cannot run in its current version , which undermines reproducibility.

---

> ### Author Rebuttal · Authors · 2025-07-31
>
> We appreciate the valuable comments by Reviewer NZTA.
>
> **Q1: The use of dual-branch architecture in CL.**
>
> **A1:** We clarify that our method differs fundamentally from dual-branch approaches in both architecture and learning dynamics. Specifically, dual-branch methods like DualNet and CLS-ER rely on buffer to preserve historical knowledge, while LEAR is Rehearsal-Free. Additionally, these methods fully fine-tune the entire pretrained models, whereas LEAR only fine-tunes the last three layers of the backbone. Furthermore, existing approaches usually lack or even omit the feature interaction between the two branches, while LEAR employs MIBPA, KLDBFA, and HSICBCO to establish robust global-local knowledge consolidation and adaptation.
>
> **Q2: The differences from the existing dynamic expandable models.**
>
> **A2:** Conventional approaches typically utilize a single pre-trained ViT as the foundational backbone to facilitate adapter integration. In contrast, the proposed LEAR framework optimizes a collaborative backbone architecture, incorporating both global and local backbones to deliver task-agnostic and adaptive feature representations. Leveraging this collaborative structure, LEAR executes expert construction via a comprehensive feature integration process. This architecture effectively accommodates heterogeneous data domains, representing a novel direction within the continual learning landscape. Furthermore, while baseline methods rely on routers or dynamic weighting schemes for expert aggregation, LEAR introduces an innovative Expert Selection Mechanism (ESM) that autonomously identifies the most suitable expert and leverages its parameter information for constructing new experts. This mechanism enhances the learning of novel tasks and constitutes an unexplored strategy in current continual learning research. To further demonstrate LEAR's effectiveness compared to other expansion-based approaches in MDCL,We have supplemented the experimental comparison between LEAR and three baseline methods: EASE[1], SEMA[2], and the LoRA-based approach InfLoRA [3], on both ETI and TRC sequences. Experimental results in Table 1 confirm LEAR's significant performance advantage over these baselines.
>
> ### Table1: Supplementary performance comparison between relevant methods and LEAR
> |Methods|EuroSAT|TImg|ImgR|CUB200|C100|MNIST|RESISC45|ChestX|C10|Disease|Avg|
> |-|-|-|-|-|-|-|-|-|-|-|-|
> |EASE|4.36|35.32|22.78|60.98|40.56|58.24|34.72|10.42|94.38|90.84|45.26|
> |InfLoRA|76.79|66.25|33.74|71.45|77.23|92.15|77.56|32.35|87.70|93.66|70.89|
> |SEMA|10.47|67.76|47.05|70.35|55.94|69.87|89.03|25.62|89.23|96.96|62.23|
> |SLCA|85.78|0.37|0.52|0.79|1.55|10.34|2.15|9.59|10.17|91.35|21.26|
> |LEAR|95.89|81.25|69.57|84.12|85.30|98.56|92.92|45.45|96.60|99.30|84.90|
>
> |Methods|TImg|RESISC45|CUB200|ChestX|ImgR|EuroSAT|MNIST|C10|Disease|C100|Avg|
> |-|-|-|-|-|-|-|-|-|-|-|-|
> |EASE|15.22|48.51|50.98|5.17|32.78|54.72|81.14|67.46|79.84|81.84|51.77|
> |InfLoRA|32.63|68.98|75.75|21.38|51.74|73.69|86.02|86.17|83.45|82.49|66.23|
> |SEMA|56.24|11.43|46.35|8.45|55.93|67.65|80.41|82.23|91.11|80.36|58.02|
> |SLCA|63.67|0.78|0.67|14.84|0.62|11.31|11.20|13.70|8.21|51.18|17.62|
> |LEAR|80.30|92.41|83.49|44.67|69.28|96.62|98.53|95.84|99.16|86.22|84.65|
>
> **Q3: The novelty of MIBPA and the motivation of MI rather than CE.**
>
> **A3:** Ensuring consistency with historical predictions has been extensively adopted in regularization-based approaches. Typically, these methods formulate the current active model (classifier) as a student module and duplicate it as a teacher module following each task transition. Catastrophic forgetting is mitigated by minimizing the divergence between the outputs of the student and teacher modules. While this framework offers robust stability, it often compromises adaptability when confronted with complex datasets. In contrast to conventional regularization-based techniques, the proposed MIBPA introduces two key innovations:
>
> (1) Rather than duplicating the entire classifier as a teacher module after each task switch, MIBPA incorporates a parameter-sharing auxiliary model that retains select parameters from the global backbone (specifically, the final three layers) and leverages this auxiliary model to guide the optimization of the global backbone. This strategy effectively reduces the overall parameter count;
>
> (2) Whereas existing methods enforce output alignment between the current classifier and the teacher module during training, MIBPA focuses on preserving the prediction consistency of each historical expert during global backbone optimization, while simultaneously encouraging the current expert to assimilate data from the ongoing task (MI is symmetric). This approach utilizes supervised signals from all previously acquired experts to control the optimization process of the global backbone and is entirely different from the existing regularization-based methodologies.
>
> Exceptionally, SLCA [4] models the class-wise distributions of historical tasks and aligning the classification layers in a post-hoc fashion. However, the experimental results in Table 1 demonstrate that this modeling and alignment approach struggles to adapt to the complex task prediction distributions in MDCL.
>
> Additionally, we performed ablation studies substituting the mutual information component in MIBPA with cross-entropy (CE) loss. The experimental results ('CBE+CE' in Table 3) demonstrate a substantial performance degradation when employing asymmetric CE. This is because the model over-emphasizes the historical prediction distributions while neglecting to learn the current task. This empirical evidence motivated our choice of mutual information.
>
> **Q4: Truly novel components in LEAR.**
>
> **A4:** We claim that the expert construction process is a known technology that was adopted in this study. In contrast, the proposed collaborative backbone structure, Mutual Information-Based Prediction Alignment, and Kullback–Leibler (KL) Divergence-Based Feature Alignment are novel technologies, which are not explored in the existing continual learning studies.
>
> **Q5: The distribution modeling in Eq7.**
>
> **A5:** We clarify that no probabilistic models were employed in this process. We sincerely invite you to refer to our response to Q1 in the rebuttal to reviewer tTuR, as we are constrained by word limits and unable to answer this question.
>
> **Q6: The task definition.**
>
> **A6:** We clarify that in the MDCL setting, each individual task corresponds to a complete, undivided benchmark dataset. Regarding the second setting you mentioned, we designed three distinct data sequences, each comprising two datasets and further partitioned into 10 tasks per dataset (20 tasks total). Each task is trained only 1 epoch. The experimental results demonstrate that LEAR still shows robust adaptability to domain shift and outperforms all three compared methods in terms of average task performance.
>
> ### Table2
> |Method|CUB-ImgR||TinyImg-ImgR||ImgR-CUB||
> |-|-|-|-|-|-|-|
> ||Avg|Last|Avg|Last|Avg|Last|
> |DER++(Re)|60.85|83.71|51.18|83.93|60.50|82.76|
> |MoEAdapter|12.61|23.53|25.60|17.57|43.83|75.85|
> |DualPrompt|52.34|67.44|60.67|42.17|57.08|79.98|
> |LEAR|78.74|75.34|72.36|69.47|77.82|81.39|
>
> **Q7: Improvement of ablation study.**
>
> **A7:** we have conducted extensive additional experiments on ETI and TRC sequences during the rebuttal phase and reorganized the ablation study tables accordingly:
>
> ### Table3: Results of revised ablation studies
> |Methods|EuroSAT|TImg|ImgR|CUB200|C100|MNIST|RESISC45|ChestX|C10|Disease|Avg|
> |-|-|-|-|-|-|-|-|-|-|-|-|
> |CBE|92.38|76.49|62.42|78.45|82.05|95.05|90.93|38.73|94.17|99.04|80.97|
> |CBE+MI|95.08|80.93|68.19|82.49|84.68|97.15|91.29|39.56|96.18|98.28|83.38|
> |CBE+CE|90.25|36.96|21.21|10.69|29.69|48.42|51.80|42.83|75.59|70.50|47.79|
> |CBE+KL|92.18|78.02|66.13|78.97|82.50|92.24|90.54|44.58|95.81|99.13|82.01|
> |CBE+HSIC|93.65|76.97|64.37|78.56|82.35|91.88|91.29|45.53|95.49|99.06|81.92|
> |CBE+MI+KL|95.35|81.61|68.30|83.56|85.54|97.43|92.28|46.44|96.22|99.01|84.57|
> |CBE+MI+HSIC|95.87|80.85|68.74|83.15|85.52|98.03|92.08|42.43|96.33|98.91|84.19|
> |CBE+KL+HSIC|93.24|78.22|66.45|80.49|82.55|92.21|91.39|45.26|95.59|99.08|82.45|
> |LEAR|95.89|81.25|69.57|84.12|85.30|98.56|92.92|45.45|96.60|99.30|84.90|
>
> |Methods|TImg|RESISC45|CUB200|ChestX|ImgR|EuroSAT|MNIST|C10|Disease|C100|Avg|
> |-|-|-|-|-|-|-|-|-|-|-|-|
> |CBE|79.01|89.72|80.68|30.26|64.06|92.34|94.15|94.44|97.05|86.19|80.79|
> |CBE+MI|81.24|90.76|82.51|40.13|67.98|95.19|97.68|95.64|98.03|85.77|83.49|
> |CBE+CE|79.60|71.17|26.80|35.65|43.22|93.35|91.79|94.81|96.78|73.19|70.64|
> |CBE+KL|78.77|90.36|80.22|40.21|66.52|94.67|97.01|95.27|98.40|87.02|82.85|
> |CBE+HSIC|79.14|90.21|80.54|42.12|66.21|93.26|94.24|94.48|97.25|85.41|82.29|
> |CBE+MI+KL|81.24|90.39|82.60|44.11|68.58|96.01|98.37|96.22|98.86|86.17|84.25|
> |CBE+MI+HSIC|82.21|90.43|83.05|44.74|68.41|95.18|95.86|96.09|98.91|85.85|84.07|
> |CBE+KL+HSIC|79.67|89.88|81.44|42.47|66.27|95.07|96.23|95.54|98.91|86.74|83.22|
> |LEAR|80.30|92.41|83.49|44.67|69.28|96.62|98.53|95.84|99.16|86.22|84.65|
>
> Where "CBE" denotes the configuration using solely the collaborative backbone with task-specific network expansion and ESM expert selection, and "CBE+MIBPA/KLDBFA/HSICBCO" represents CBE augmented with individual components or combination of components. The results demonstrate that each regularization term contributes to varying degrees of performance improvement over CBE across both ETI and TRC.
>
> **Q8: Code implement.**
>
> **A8:** Should our paper be accepted, we will definitely release the full codebase with comprehensive documentation.
>
> ### References:
>
> [1]Expandable subspace ensemble for pre-trained model-based class-incremental learning//CVPR24
>
> [2]Self-Expansion of Pre-trained Models with Mixture of Adapters for Continual Learning//CVPR25
>
> [3]Inflora: Interference-free low-rank adaptation for continual learning//CVPR24
>
> [4]SLCA: Slow Learner with Classifier Alignment for Continual Learning on a Pre-trained Model//ICCV23

---

> > ### Comment · Reviewer_NZTA · 2025-08-05
> >
> > I will maintain my original score. I believe that this work requires substantial revisions before it is ready for publication. In particular, the paper should place more emphasis on the truly novel components and reduce the focus on the more straightforward applications of existing techniques.

---

> ### Author Response · Authors · 2025-08-05
> **Fundamental Differences Between LEAR and Related Approaches**
>
> Thank you for your thoughtful feedback and for taking the time to review our paper. The proposed LEAR diverges fundamentally from existing techniques as follows.
>
> **Distinctiveness from Dual-Branch Methods**
>
> Compared to existing dual-branch methods like DualNet [1],TwF [2] and CLS-ER [3], LEAR diverges significantly in three aspects:
> - **Rehearsal-Free**: Conventional dual-branch approaches often rely on memorized buffers​​ to retain historical knowledge. In contrast, the dual-backbone structure in the proposed LEAR does not require accessing any past examples.
>
> - **Optimization Scope**: Unlike the existing methods that optimizes all parameters of the backbone, resulting in huge computational costs. In contrast, the proposed LEAR aims to optimize a few high-level representation layers of the backbones, which not only reduce computational cost but also improve the model's plasticity.
>
> - **Knowledge Interaction**: Current dual-branch methods often lack feature interaction mechanisms, whereas LEAR incorporates three novel mechanisms (MIBPA, KLDBFA, HSICBCO) to enable robust global-local adaptation.
>
> **Differentiation from Expansion-based Paradigms**
>
> Existing expansion approaches fall into three categories:
> - **Backbone-wise**: DER [4], Foster [5], MEMO [6], BEEF [7] train separate backbones per task
> - **Token-wise**: Dytox [8], L2P [9], DualPrompt [10], CODA-Prompt [11] utilize dynamic tokens
> - **Expert-wise**: MoE-Adapters [12], EASE [13], SEMA [14] grow task-specific subnetworks
>
> LEAR belongs to the third category and introduces **two fundamental innovations**:
>
> - **Collaborative Backbone Architecture**: Unlike  adapter-based expansion approaches with a single, fully frozen backbone, LEAR fine-tunes the last three layers of both global and local backbones to deliver task-agnostic and adaptive feature representations to heterogeneous data domains.
>
> - **Autonomous Expert Expansion and Selection**: While existing methods use routers for expert aggregation, this study introduces a novel Expert Selection Mechanism (ESM) that automatically determines an optimal expert and transfers parametric knowledge of the selected expert into the new expert construction process, which is an unexplored strategy in continual learning.
>
> These architectural and algorithmic innovations position LEAR as a novel direction in domain-incremental and rehearsal-free continual learning, advancing beyond incremental adaptations of existing techniques.
>
> ### References:
>
> [1] DualNet: Continual Learning, Fast and Slow//NeurIPS21
>
> [2] Transfer without forgetting//ECCV22
>
> [3] Learning fast, learning slow: A general continual learning method based on complementary learning system//ICLR22
>
> [4] Der: Dynamically expandable representation for class incremental learning//CVPR21
>
> [5] Foster: Feature boosting and compression for class-incremental learning//ECCV22
>
> [6] A model or 603 exemplars: Towards memory-efficient class-incremental learning//ICLR23
>
> [7] Beef: Bi-compatible class-incremental learning via energy-based expansion and fusion/ICLR22
>
> [8] Dytox: Transformers for continual learning with dynamic token expansion//CVPR22
>
> [9] Learning to prompt for continual learning//CVPR22
>
> [10] Dualprompt: Complementary prompting for rehearsal-free continual learning//ECCV22
>
> [11] Coda-prompt: Continual decomposed attention-based prompting for rehearsal-free continual learning//CVPR23
>
> [12] Boosting continual learning of vision-language models via mixture-of-experts adapters//CVPR24
>
> [13] Expandable subspace ensemble for pre-trained model-based class-incremental learning//CVPR24
>
> [14] Self-Expansion of Pre-trained Models with Mixture of Adapters for Continual Learning//CVPR25

---

> ### Author Response · Authors · 2025-08-06
> **Revised Introduction (Part 1)**
>
> We sincerely appreciate your time and valuable comments again.
>
> Incorporating feedback from all reviewers, we have restructured the **Introduction section** to **emphasize LEAR's novel contributions** in the next comment.
>
> Specifically, through systematic analysis of **three key challenges (plasticity, stability and efficiency)** in the proposed MDCL scenario (in contrast to CIL and conventional DIL), we demonstrate:
>
> (1) The fundamental reason for existing continual learning methods' inability to address MDCL,
>
> (2) Present our corresponding solutions for each challenge,
>
> and ultimately forming the comprehensive LEAR framework to address the MDCL.
>
> Meanwhile, we have incorporated the **enhanced ablation studies** from the rebuttal and their analyses into the main text to explicitly demonstrate the contributions of each module in LEAR and their various combinations.
>
> We sincerely hope our revisions meet your expectations.
>
> **References**
>
> [1] S-prompts learning with pre-trained transformers//neurips22
>
> [2] Preventing zero-shot transfer degradation in continual learning of vision-language models//ICCV23
>
> [3] Coleclip: Open-domain continual learning via joint task prompt and vocabulary learning//TNNLS25
>
> [4] DualNet: Continual Learning, Fast and Slow//NeurIPS21
>
> [5] Transfer without forgetting//ECCV22
>
> [6] Learning fast, learning slow: A general continual learning method based on complementary learning system//ICLR22
>
> [7] Dualprompt: Complementary prompting for rehearsal-free continual learning//ECCV22
>
> [8] Boosting continual learning of vision-language models via mixture-of-experts adapters//CVPR24
>
> [9] Expandable subspace ensemble for pre-trained model-based class-incremental learning//CVPR24
>
> [10] Self-Expansion of Pre-trained Models with Mixture of Adapters for Continual Learning//CVPR25

---

> ### Author Response · Authors · 2025-08-06
> **Revised Introduction (Part 2)**
>
> Current Continual Learning (CL) research primarily focuses on Class-Incremental Learning (CIL) within a single domain, neglecting the scenario of learning across multiple domains, known as Domain-Incremental Learning (DIL). Although studies [1-3] have investigated DIL, their evaluated domains (e.g., Aircraft, MNIST) have achieved near-perfect accuracy with pre-trained ViTs, making these benchmarks inadequate for assessing genuine continual learning capabilities. Therefore, we establish a more challenging and more realistic Multi-domain Continual learning (MDCL) scenario in which the discrepancy among tasks remains large. In this study, we aim to improve the model's performance in MDCL by considering three aspects including **plasticity, stability and efficiency**. To implement this goal, we propose a novel approach called LEAR and its core idea is to fully explore the stable and dynamic representations extracted by the pre-trained ViT backbones to achieve fast adaptation while adaptively optimizing the backbones to maintain all previously learned information.
>
> **(1) Plasticity.** Existing dual-branch approaches [4-6] and Expansion-Based Methods (EBMs) [7-10] improve downstream task performance by integrating task-specific prompts or adapters into a fixed pretrained backbone. However, these methods focus on exploring representations from a single pre-trained backbone, which fails to address more challenging data domains such as CUB200. Thus, to improve plasticity in a challenging MDCL scenario, we introduce a novel collaborative backbone architecture for LEAR, comprising a global and a local backbone, designed to capture general and task-specific information across all tasks. Leveraging this collaborative backbone structure, the proposed LEAR framework dynamically generates a lightweight expert to learn the decision boundary for each new task, thereby achieving commendable performance. The results presented in Tab. 1 and 2 of the paper demonstrate that our method achieves superior performance on most individual datasets in the MDCL scenario, which also validates that EBMs with frozen pretrained backbone cannot provide sufficient plasticity in MDCL.
>
> **(2) Stability.** Many Pre-Trained Models (PTMs) based methods have shown to achieve excellent stability in continual learning by dynamically creating new sub-models. However, the excellent stability is usually achieved by freezing all parameters of the PTMs during the training, which prevents from learning new tasks effectively, especially when facing the severe domain shifts (ChestX→ImageNet-R) in long task sequences. To address the limitation of the existing PTMs-based methods, we propose a unified optimization function to regulate the optimization behaviour of the LEAR framework. This function consists of a MIBPA loss and a KLDBFA loss. The former dynamically optimizes the global backbone while preventing negative knowledge transfer at the prediction level, and the latter aligns historical and current representation distributions at the feature level. Such a design enables LEAR to achieve rehearsal-free continual learning by actively consolidating historical knowledge at both the prediction and feature levels when fine-tuning the collaborative backbones with new task data, rather than simply freezing parameters passively. Such a design has not been explored in the existing CL field. Furthermore, to mitigate optimization interference and information redundancy between the collaborative backbones, we propose a novel Hilbert-Schmidt Independence Criterion (HSIC)-Based Collaborative Optimization (HSICBCO) strategy to encourage two backbones to capture different semantic information, thus promoting effective complementary learning of MDCL tasks. The experimental results demonstrate that LEAR significantly outperforms all baseline methods in terms of overall average accuracy in MDCL scenarios.
>
> **(3) Efficiency.** Many existing expansion-based methods usually ignore the task relevance and do not explore the previously learned parameter information to accelerate the new task learning. As a result, these methods optimize each new expert from scratch, resulting in leading to considerable computational costs and parameter redundancy. To address this issue, we aim to promote the efficient learning process of LEAR by proposing a novel Expert Selection Mechanism (ESM) that selectively transfers the parameter information learned by a selected expert into the new expert construction process. Specifically, the proposed ESM models each expert's knowledge as a Gaussian memory distribution and only preserve its critical statistical information. For each new task, the proposed ESM selects the most relevant expert by minimizing the Mahalanobis distance between stored distributions and incoming data, and reuses its parameters to facilitate new task learning. During testing phase, ESM autonomously routing testing samples to the most suitable expert in a task-agnostic manner.

---

### Official Review · Reviewer_WD1L · 2025-06-28

**Clarity:** 1
**Significance:** 3
**Originality:** 3
**Rating:** 4
**Confidence:** 3

**Summary:**

The paper addresses the limitations of existing class-incremental learning (CIL) methods, which often operate within a single domain and fail to reflect real-world scenarios. To overcome this, the authors introduce a new Multi-Domain Continual Learning (MDCL) setting, where each task corresponds to a distinct dataset, introducing both class and domain shift. To tackle MDCL, they propose a collaborative architecture consisting of a global backbone and a local backbone to capture general and task-specific information. To enhance performance and reduce forgetting, the approach integrates several components such as Mutual Information-Based Prediction Alignment (MIBPA), Kullback–Leibler Divergence-Based Feature Alignment (KLDBFA), Hilbert-Schmidt Independence Criterion-Based Collaborative Optimization (HSICCO), and Expert Selection Mechanism (ESM).

**Questions:**

Questions:
1. Could you clarify what is stored for each task after training, specifically regarding the parts of the local backbone that need to be stored?
2. What would be the total number of parameters of this pipeline, not just the trainable parameters, but all the parameters that need to be stored, including those from the experts, task-specific covariances, etc.?
3. Could you please provide the performance of the collaborative architecture with the global and local backbones and without any other components?
4. Could you also provide the performance on standard single-domain CIL benchmarks for comparison, such as CIFAR-100, CUB, or ImageNet-R with 10-task splits?

**Ethical Concerns:**

["NO or VERY MINOR ethics concerns only"]

**Final Justification:**

I sincerely thank the authors for their rebuttal. After carefully considering the other reviews and the rebuttal, I believe this is a good paper that outperforms many strong baselines. However, the presentation of the method was not very clear, making it difficult to fully understand how the approach works or achieves these results. The additional ablation study provided in the rebuttal partially addressed my concerns, and I have decided to raise my score. I still encourage the authors to further improve the writing and presentation of the method in order to maximize the impact of this work.

**Limitations:**

Yes, but a more thorough analysis of the number of parameters could have been provided.

**Paper Formatting Concerns:**

No Formatting Issues

**Quality:**

2

**Strengths And Weaknesses:**

Strengths:
1. The MDCL setting that the paper presents more accurately represents the real-world CL scenario and can introduce a more challenging benchmark for the community to pursue, given many CIL methods already achieve commendable results on a single domain.
2. The experiments show that many current methods fail to handle MDCL, yet the proposed method shows great performance, surpassing state-of-the-art baselines.

Weaknesses:
1. The paper is hard to follow due to the overwhelming introduction of multiple components without sufficient context. It would be clearer if the authors provided more in-depth discussion before each component explaining the motivation. For example, for mitigating forgetting, the authors say they introduce KLDBFA—it would have been better to briefly reference options like regularization and distillation. By making this connection to previous work, they could make it easier to understand and then mention why they decided to go with KL. The same approach could have been applied to each component.
2.  Before the MDCL setting, it would be better to also compare the proposed approach on the standard single-domain CIL setting. The proposed method should also work on a single domain, so having that comparison helps establish the method’s performance, as all literature reports results in that scenario. It would be easier to compare, and it also strengthens the work by showing it can handle a single domain. You could then move to MDCL to show that current methods fail on that setting.
3. The ablation studies on different model components should be more thorough. There is w/o MI, KL, and HSIC, which show method performance without each of these components. However, ESM is not discussed (there is a visualization in Table 2b, but accuracy without ESM is not given). Additionally, HSIC in Table 2 shows no meaningful improvement—the model is only 0.11% worse when removing it, which is not statistically significant, so given its complexity, its inclusion may not be justified. The results reported in Table 2 are much better than any other baselines, but it is unclear where these improvements originate from. To provide more insight, the authors should first present results for the collaborative architecture with only the global and local backbones, then progressively add a component or a combination of components to demonstrate their contributions and possible interactions. Having the joint fine-tuning results, where the model is fine-tuned in a non-continual setting on all the data, would also serve as a valuable upper bound for comparison.

I would reconsider my evaluation if my concerns regarding the experiments are addressed. I would also like to review the comments from other reviewers.

---

> ### Author Rebuttal · Authors · 2025-07-31
>
> We sincerely appreciate the reviewer's constructive comments and recognition of the (1) the practical significance of our proposed Multi-Domain Continual Learning (MDCL) compared to conventional CIL learning settings, and (2) the demonstrated superior performance of LEAR over prior methods in this challenging scenario.
>
> For weaknesses and questions, we provide our feedback as follows.
>
> **Q1: The discussion of motivation before introduction of components.**
>
> **A1:** We sincerely appreciate your invaluable suggestions and guidance. Due to space constraints in the main text, it's challenging to provide a detailed elaboration on the motivation of each module. We have thoroughly revised the introduction of the KLDBFA module to provide clearer motivation and methodological justification. The revised content is presented as follows:
>
> The iterative updating of the pre-trained backbones facilitates the temporal capture of local representations, thereby potentially enhancing the acquisition of novel tasks. However, this process risks inducing adverse knowledge transfer and performance degradation across historical experts. Regularization methods like EWC [1] and MAS [2], which typically impose constraints on parameter updates, are not desirable to capture the complex distributional shifts across domains, while knowledge distillation methods like LWF [3] and iCaRL [4] require maintaining additional teacher networks that become computationally prohibitive as the number of tasks grows. To address these limitations, we propose Kullback-Leibler Divergence-Based Feature Alignment (KLDBFA), designed to preserve crucial historical parameters during the optimization of the local backbone.
>
> Our design rationale for selecting KL divergence stems from two key considerations: Firstly, modern generative evaluation metrics (e.g., FID, Kernel MMD) operate on the Gaussian distribution assumption in high-dimensional feature spaces. This motivates us to model backbone features as Gaussian distributions. Fig. 1 in Appendix-C from the SM also provides additional empirical validation for the Gaussian distribution assumption. Secondly, KL divergence offers unique advantages over alternative distributional metrics: (1) Its directional property enables targeted constraint of current features toward historical distributions, unlike symmetric metrics (e.g., Jensen-Shannon divergence); (2) It maintains computational efficiency compared to expensive metrics like MMD or Wasserstein distance. These characteristics make KL divergence ideally suited for continual learning scenarios requiring efficient knowledge preservation.
>
> **Q2: Performance comparison on CIL settings.**
>
> **A2:** We sincerely agree with your comment. As MDCL is indeed significantly more challenging than standard CIL, comparing with existing methods under the CIL setting would further demonstrate and strengthen the superiority of LEAR. Therefore, we have further supplemented the comparison of LEAR with relevant methods on CIL performance over CIFAR-100, CUB-200, ImageNet-R, and TinyImageNet, each partitioned into 10 tasks:
> ### Table 1: Results of CIL
> |Method|C100|CUB|ImgR|TinyImg|
> |-|-|-|-|-|
> |RanPAC|92.23|**90.32**|78.11|72.89|
> |MoE|85.21|82.26|76.77|80.23|
> |L2P|82.76|79.23|73.73|76.37|
> |DualPrompt|84.12|83.21|**78.47**|81.38|
> |CODAPrompt|86.33|83.36|74.45|82.80|
> |LEAR|**95.80**|88.38|77.67|**85.86**|
>
> The results demonstrate LEAR's excellent adaptability to CIL settings, where it achieves competitive performance. We have incorporated this comparative analysis, along with detailed discussions, prior to presenting the MDCL results in the revised paper.
>
> **Q3: Suggestions of ablation studies and performance upper bound.**
>
> **A3:** We note that all four reviewers raise concerns in the individual contributions of LEAR's components and have provided consistent suggestions for improving our ablation studies. Recognizing the importance of this issue, we have conducted extensive additional experiments on ETI and TRC sequences during the rebuttal phase and reorganized the ablation study tables accordingly:
> ### Table 2: Results of ablation studies
> |Methods|EuroSAT|TImg|ImgR|CUB200|C100|MNIST|RESISC45|ChestX|C10|Disease|Avg|
> |-|-|-|-|-|-|-|-|-|-|-|-|
> |UpperBound|96.71|82.77|72.81|85.08|86.39|98.97|93.72|50.14|96.49|99.35|86.24|
> |CB|19.59|8.37|11.81|12.29|36.92|49.88|61.14|30.04|92.79|99.00|42.18|
> |CBE|92.38|76.49|62.42|78.45|82.05|95.05|90.93|38.73|94.17|99.04|80.97|
> |CBE+MI|95.08|80.93|68.19|82.49|84.68|97.15|91.29|39.56|96.18|98.28|83.38|
> |CBE+KL|92.18|78.02|66.13|78.97|82.50|92.24|90.54|44.58|95.81|99.13|82.01|
> |CBE+HSIC|93.65|76.97|64.37|78.56|82.35|91.88|91.29|45.53|95.49|99.06|81.92|
> |CBE+MI+KL|95.35|81.61|68.30|83.56|85.54|97.43|92.28|46.44|96.22|99.01|84.57|
> |CBE+MI+HSIC|95.87|80.85|68.74|83.15|85.52|98.03|92.08|42.43|96.33|98.91|84.19|
> |CBE+KL+HSIC|93.24|78.22|66.45|80.49|82.55|92.21|91.39|45.26|95.59|99.08|82.45|
> |LEAR|95.89|81.25|69.57|84.12|85.30|98.56|92.92|45.45|96.60|99.30|84.90|
> |LEAR w/o ESM|3.72|1.25|2.36|83.95|8.24|4.91|14.63|1.05|17.85|99.15|23.71|
>
>
> |Methods|TImg|RESISC45|CUB200|ChestX|ImgR|EuroSAT|MNIST|C10|Disease|C100|Avg|
> |-|-|-|-|-|-|-|-|-|-|-|-|
> |CB|8.43|22.67|39.45|23.51|25.67|42.74|60.95|64.50|93.45|84.87|46.62|
> |CBE|79.01|89.72|80.68|30.26|64.06|92.34|94.15|94.44|97.05|86.19|80.79|
> |CBE+MI|81.24|90.76|82.51|40.13|67.98|95.19|97.68|95.64|98.03|85.77|83.49|
> |CBE+KL|78.77|90.36|80.22|40.21|66.52|94.67|97.01|95.27|98.40|87.02|82.85|
> |CBE+HSIC|79.14|90.21|80.54|42.12|66.21|93.26|94.24|94.48|97.25|85.41|82.29|
> |CBE+MI+KL|81.24|90.39|82.60|44.11|68.58|96.01|98.37|96.22|98.86|86.17|84.25|
> |CBE+MI+HSIC|82.21|90.43|83.05|44.74|68.41|95.18|95.86|96.09|98.91|85.85|84.07|
> |CBE+KL+HSIC|79.67|89.88|81.44|42.47|66.27|95.07|96.23|95.54|98.91|86.74|83.22|
> |LEAR|80.30|92.41|83.49|44.67|69.28|96.62|98.53|95.84|99.16|86.22|84.65|
> |LEAR w/o ESM|4.27|4.83|68.49|9.42|3.76|14.59|1.98|6.33|1.64|9.25|12.46|
>
> Where ''CB'' denotes using only the collaborative backbone with a single shared expert network across all datasets; ''CBE'' extends CB with task-specific expert network expansion and ESM expert selection; and "CBE+MI/KL/HSIC" represents CBE augmented with individual components or a combination of components. ''UpperBound'' denotes the results in a non-continual setting.
>
> This table demonstrates that the dynamically expanded expert network ('CB'->'CBE') significantly enhances model performance on both ETI and TRC. Furthermore, each regularization component and their respective combinations yield distinct enhancements in performance relative to CBE across both sequences.
>
>
> Furthermore, the ablation study for ESM was initially omitted because ESM serves as the core component of LEAR. Without ESM, LEAR would lose its capability to dynamically select appropriate experts for domain-specific evaluation under task-agnostic conditions. To address this omission, we further conduct experiments with randomly selected experts during the beginning and testing phase of each task, which is denoted as ''LEAR w/o ESM''. The results demonstrate that the model's performance will significantly drop when inappropriate experts are chosen, thereby demonstrating the necessity of our ESM.
>
> **Q4: Storage at the end of the task.**
>
> **A4:** Following the network's input-output pipeline, we preserve the following components per task: (1) The last three layers of both global and local backbones, which are saved and frozen as Frozen-Global and Frozen-Local backbones (overwrite their corresponding components saved from the previous task); (2) Task-specific mean vectors and covariance matrices which computed from the features extracted by the Static backbone for a random subset of training samples.
>
> **Q5: Total parameter count.**
>
> **A5:** The total parameter count reaches approximately 241.7M, with the following detailed composition: (1) Two complete ViT backbones (Global and Local) comprise 172M (86M\*2) parameters; (2) Three backbone variants (Frozen-Global, Frozen-Local, and Static) collectively contribute 63.6M (21.2M\*3) parameters; (3) 10 task-specific expert networks account for about 4.8M (0.38M\*10 for 10 fc layers + 1M for all classifiers) parameters; and (4) Mean vectors and covariance matrices for 10 tasks sum to 1.3M (0.13M\* 10 for 10 mu \& sigma) parameters.
>
> **Q6: Performance of solely collaborative architecture.**
>
> **A6:** Please refer to the results of ''CBE'' in Table 2. However, if you are referring to the results after excluding the task-specific expert network expansion, please refer to those associated with ''CB''.
>
> **Q7: Performance on CIL setting.**
>
> **A7:** Please refer to ''**A2**''.
>
> ### References:
>
> [1] Overcoming catastrophic forgetting in neural networks//PNAS17
>
> [2] Task-free continual learning//CVPR19
>
> [3] Learning without forgetting//TPAMI17
>
> [4] iCaRL: Incremental Classifier and Representation Learning//CVPR17

---

> ### Comment · Reviewer_WD1L · 2025-08-05
>
> I thank the authors for their detailed rebuttal. It helped clarify several points, and in particular, addressed my concerns regarding the ablation study. I appreciate the effort in responding comprehensively.
>
> One suggestion I would like to share is that the paper currently includes a number of components—each contributing incrementally to the final performance. While these additions are reasonable, the overall structure feels a bit too complex for a conference paper. As a result, it was difficult to identify a single core idea or takeaway that clearly distinguishes this work from prior approaches.
>
> In general, successful conference papers tend to center around a focused contribution that addresses a well-defined limitation of existing methods. In this case, although the individual components have been applied in a slightly different way, many of them build on existing techniques and the combined architecture may come across as a collection of ideas rather than a unified, novel contribution. This makes it harder to articulate the main innovation when reading the paper.
>
> Of course, this is a matter of personal preference, and I do not hold it against the authors in my final assessment. I have updated my score accordingly, as I still believe the paper presents valuable insights and shows strong empirical results. I hope this feedback is helpful, and I encourage the authors to consider a more streamlined presentation.

---

> ### Author Response · Authors · 2025-08-06
> **Revised Introduction (Part 1)**
>
> We sincerely appreciate the reviewer's insightful feedback and recognition of our work's value. We are honored that our rebuttal addressed the concerns raised.
>
> Incorporating feedback from all reviewers, in the next comment we have restructured the **Introduction section** to **better articulate the core contributions while explicitly differentiating our approach from existing methods, thereby enhancing readability.**
>
> Specifically, through systematic analysis of **three key challenges (plasticity, stability and efficiency)** in the proposed MDCL scenario (in contrast to CIL and conventional DIL), we demonstrate:
>
> (1) The fundamental reason for existing continual learning methods' inability to address MDCL,
>
> (2) Present our corresponding solutions for each challenge,
>
> and ultimately forming the comprehensive LEAR framework to address the MDCL.
>
> Meanwhile, we have incorporated the **enhanced ablation studies** from the rebuttal and their analyses into the main text to explicitly demonstrate the contributions of each module in LEAR and their various combinations.
>
> We sincerely hope our revisions meet your expectations.
>
> **References**
>
> [1] S-prompts learning with pre-trained transformers//neurips22
>
> [2] Preventing zero-shot transfer degradation in continual learning of vision-language models//ICCV23
>
> [3] Coleclip: Open-domain continual learning via joint task prompt and vocabulary learning//TNNLS25
>
> [4] DualNet: Continual Learning, Fast and Slow//NeurIPS21
>
> [5] Transfer without forgetting//ECCV22
>
> [6] Learning fast, learning slow: A general continual learning method based on complementary learning system//ICLR22
>
> [7] Dualprompt: Complementary prompting for rehearsal-free continual learning//ECCV22
>
> [8] Boosting continual learning of vision-language models via mixture-of-experts adapters//CVPR24
>
> [9] Expandable subspace ensemble for pre-trained model-based class-incremental learning//CVPR24
>
> [10] Self-Expansion of Pre-trained Models with Mixture of Adapters for Continual Learning//CVPR25

---

> ### Author Response · Authors · 2025-08-06
> **Revised Introduction (Part 2)**
>
> Current Continual Learning (CL) research primarily focuses on Class-Incremental Learning (CIL) within a single domain, neglecting the scenario of learning across multiple domains, known as Domain-Incremental Learning (DIL). Although studies [1-3] have investigated DIL, their evaluated domains (e.g., Aircraft, MNIST) have achieved near-perfect accuracy with pre-trained ViTs, making these benchmarks inadequate for assessing genuine continual learning capabilities. Therefore, we establish a more challenging and more realistic Multi-domain Continual learning (MDCL) scenario in which the discrepancy among tasks remains large. In this study, we aim to improve the model's performance in MDCL by considering three aspects including **plasticity, stability and efficiency**. To implement this goal, we propose a novel approach called LEAR and its core idea is to fully explore the stable and dynamic representations extracted by the pre-trained ViT backbones to achieve fast adaptation while adaptively optimizing the backbones to maintain all previously learned information.
>
> **(1) Plasticity.** Existing dual-branch approaches [4-6] and Expansion-Based Methods (EBMs) [7-10] improve downstream task performance by integrating task-specific prompts or adapters into a fixed pretrained backbone. However, these methods focus on exploring representations from a single pre-trained backbone, which fails to address more challenging data domains such as CUB200. Thus, to improve plasticity in a challenging MDCL scenario, we introduce a novel collaborative backbone architecture for LEAR, comprising a global and a local backbone, designed to capture general and task-specific information across all tasks. Leveraging this collaborative backbone structure, the proposed LEAR framework dynamically generates a lightweight expert to learn the decision boundary for each new task, thereby achieving commendable performance. The results presented in Tab. 1 and 2 of the paper demonstrate that our method achieves superior performance on most individual datasets in the MDCL scenario, which also validates that EBMs with frozen pretrained backbone cannot provide sufficient plasticity in MDCL.
>
> **(2) Stability.** Many Pre-Trained Models (PTMs) based methods have shown to achieve excellent stability in continual learning by dynamically creating new sub-models. However, the excellent stability is usually achieved by freezing all parameters of the PTMs during the training, which prevents from learning new tasks effectively, especially when facing the severe domain shifts (ChestX→ImageNet-R) in long task sequences. To address the limitation of the existing PTMs-based methods, we propose a unified optimization function to regulate the optimization behaviour of the LEAR framework. This function consists of a MIBPA loss and a KLDBFA loss. The former dynamically optimizes the global backbone while preventing negative knowledge transfer at the prediction level, and the latter aligns historical and current representation distributions at the feature level. Such a design enables LEAR to achieve rehearsal-free continual learning by actively consolidating historical knowledge at both the prediction and feature levels when fine-tuning the collaborative backbones with new task data, rather than simply freezing parameters passively. Such a design has not been explored in the existing CL field. Furthermore, to mitigate optimization interference and information redundancy between the collaborative backbones, we propose a novel Hilbert-Schmidt Independence Criterion (HSIC)-Based Collaborative Optimization (HSICBCO) strategy to encourage two backbones to capture different semantic information, thus promoting effective complementary learning of MDCL tasks. The experimental results demonstrate that LEAR significantly outperforms all baseline methods in terms of overall average accuracy in MDCL scenarios.
>
> **(3) Efficiency.** Many existing expansion-based methods usually ignore the task relevance and do not explore the previously learned parameter information to accelerate the new task learning. As a result, these methods optimize each new expert from scratch, resulting in leading to considerable computational costs and parameter redundancy. To address this issue, we aim to promote the efficient learning process of LEAR by proposing a novel Expert Selection Mechanism (ESM) that selectively transfers the parameter information learned by a selected expert into the new expert construction process. Specifically, the proposed ESM models each expert's knowledge as a Gaussian memory distribution and only preserve its critical statistical information. For each new task, the proposed ESM selects the most relevant expert by minimizing the Mahalanobis distance between stored distributions and incoming data, and reuses its parameters to facilitate new task learning. During testing phase, ESM autonomously routing testing samples to the most suitable expert in a task-agnostic manner.

---

### Official Review · Reviewer_a1tB · 2025-07-02

**Clarity:** 2
**Significance:** 3
**Originality:** 3
**Rating:** 4
**Confidence:** 5

**Summary:**

The paper introduces LEAR, a buffer-free framework for multi-domain continual learning (MDCL). Specifically, to prevent forgetting while adapting to new tasks, LEAR (i) matches old vs new expert logits with a mutual-info loss (MIBPA) for the global side, (ii) constrains drift in the local feature space with a KL-divergence loss (KL-DBFA), and (iii) pushes the two branches to carry different feature representations with an HSIC independence loss. After each task, it stores only the mean vector and covariance matrix of that expert’s transformed features, computed in a fully frozen ViT feature space; when the next task arrives, it picks the closest old expert by Mahalanobis distance and clones its weights to start learning from a good initial point. Results show the proposed LEAR boosts average accuracy and decreases forgetting on three challenging domain-shuffled image benchmarks compared to prior works.

**Questions:**

1.My primary concern is that it remains unclear why the proposed method achieves such significant accuracy improvements over prior continual learning approaches.

1.1 Although the authors provide an ablation study examining the three designed regularization terms, it is difficult to discern their individual contributions. In particular, the feature separation term (HSIC) does not appear to yield a meaningful accuracy gain.

1.2 Rather than focusing solely on the regularization terms, it would be more informative to demonstrate the effectiveness of the collaborative backbone design, which is the key contribution of this work.

1.3 It is not well explained why the proposed MIBPA approach mitigates catastrophic forgetting, nor how effectively it accomplishes this.

2.In line 175, what does the "concentration operation" stand for?

3.Which layers are frozen in the auxiliary model? Lines 185–186 state that the auxiliary model freezes the final three layers of the global backbone. However, it appears that in the rest of the backbone, all layers except the last three are frozen. Could you clarify this point? Additionally, why are the last three layers chosen specifically, rather than a different number of layers?

4.What is the exact model architecture of the lightweight expert?

**Ethical Concerns:**

["NO or VERY MINOR ethics concerns only"]

**Final Justification:**

The authors have addressed most of my concerns by 1) justifying the proposed method compared to prior CL methods, and 2) conducting ablation studies to show the effectiveness of each proposed component. However, I believe the paper still requires significant revisions to clarify the differences from prior works better and to demonstrate the effectiveness of each proposed component more clearly. In this case, I maintain my current score.

**Limitations:**

Yes

**Quality:**

3

**Strengths And Weaknesses:**

Strengths:
1. This work addresses a more practical and challenging scenario, multi-domain continual learning, as opposed to the class-incremental learning that most prior methods have focused on.

2. The experimental results demonstrate significant accuracy improvements over existing methods.

Weakness:
1. The quality of the writing needs improvement. The paper makes it difficult to grasp the overall continual learning process, and Figure 1 is hard to understand until reaching the final subsection (Section 3.7) of the Methodology.
2. While the proposed method achieves notable accuracy gains over previous approaches, the underlying reasons for its success remain unclear. Please refer to the detailed comments under “Questions” below.

---

> ### Author Rebuttal · Authors · 2025-07-31
>
> **We appreciate the valuable comments by Reviewer a1tB.**
>
> **Q1: Concerns regarding writing quality.**
>
> **A1:** We have enhanced the paper's readability and improved Fig.1 to more clearly depict the main process of LEAR.
>
> **Q2: Reason for significant accuracy improvements.**
>
> **A2:** The proposed LEAR have two primary advantages over the existing continual learning methods:
>
> (1) By emulating the brain's memory structure through CLS theory, the proposed LEAR manages a collaborative backbone architecture to provide more adaptive and semantically rich representation information when compared to the existing continual learning methods, including L2P, DAP, DualPrompt, MoEAdapter that usually employ a single fixed backbone network.
>
> (2) Compared to the existing continual learning methods, the proposed LEAR integrates three mechanisms, including the Mutual Information-Based Prediction Alignment (MIBPA), Kullback–Leibler (KL) Divergence-Based Feature Alignment (KLDBFA) and HSIC-Based Collaborative Optimization (HSICBCO), aiming to maintain a good plasticity-stability trade-off and reduce information redundancy. Such a design can deal with a long sequence of complex tasks and achieve excellent performance. We evaluate the performance of the proposed approach by introducing a more challenging Multi-Domain Continual Learning (MDCL) setting that incorporates not only highly difficult datasets (such as ImageNet-R, TinyImageNet, and ChestX) but also features scenarios with both similar feature distributions between datasets (e.g., EuroSAT and RESISC45) and severe domain shifts (e.g., CIFAR100→CropDisease), as well as fine-grained datasets (CUB200) with extended task sequences. Consequently, as evidenced in the forgetting curves in Fig. 2 of Appendix-D.1 from the Supplementary Materials (SM), many baseline methods exhibit significant performance degradation in later stages of the task sequence, demonstrating their inability to learn and capture complex datasets.
>
> **Q3: Individual contributions of three regularization terms, particularly HSICBCO.**
>
> **A3:** To better demonstrate the individual contributions of each module in LEAR, we conducted additional ablation studies on two dataset sequences (ETI and TRC), as presented in the following tables:
> ### Table 1: Results of ablation studies on ETI and TRC
> |Methods|EuroSAT|TImg|ImgR|CUB200|C100|MNIST|RESISC45|ChestX|C10|Disease|Avg|Var|
> |-|-|-|-|-|-|-|-|-|-|-|-|-|
> |CBE|92.38|76.49|62.42|78.45|82.05|95.05|90.93|38.73|94.17|99.04|80.97|308
> |CBE+MIBPA|95.08|80.93|68.19|82.49|84.68|97.15|91.29|39.56|96.18|98.28|83.38|293
> |CBE+KLDBFA|92.18|78.02|66.13|78.97|82.50|92.24|90.54|44.58|95.81|99.13|82.01|244
> |CBE+HSICBCO|93.65|76.97|64.37|78.56|82.35|91.88|91.29|45.53|95.49|99.06|81.92|247
>
> |Methods|TImg|RESISC45|CUB200|ChestX|ImgR|EuroSAT|MNIST|C10|Disease|C100|Avg|Var|
> |-|-|-|-|-|-|-|-|-|-|-|-|-|
> |CBE|79.01|89.72|80.68|30.26|64.06|92.34|94.15|94.44|97.05|86.19|80.79|370
> |CBE+MIBPA|81.24|90.76|82.51|40.13|67.98|95.19|97.68|95.64|98.03|85.77|83.49|288
> |CBE+KLDBFA|78.77|90.36|80.22|40.21|66.52|94.67|97.01|95.27|98.40|87.02|82.85|291
> |CBE+HSICBCO|79.14|90.21|80.54|42.12|66.21|93.26|94.24|94.48|97.25|85.41|82.29|259
>
> Where "CBE" denotes the configuration using solely the collaborative backbone with task-specific network expansion and ESM expert selection, and "CBE+MIBPA/KLDBFA/HSICBCO" represents CBE augmented with individual components. The results demonstrate that each regularization term contributes to varying degrees of performance improvement over CBE across both ETI and TRC.
>
> Specifically, the contribution of HSICBCO can be further reflected through the variation of accuracy.  As shown in the "Var" columns of Table 1, "CBE+HSICBCO" achieves significantly reduced accuracy variance compared to "CBE" while maintaining improved average accuracy (Avg). This demonstrates that HSICBCO narrows the performance gaps across datasets, and enhances the performance on challenging datasets (e.g., ChestX, ImgR), which collectively validates HSICBCO's contribution to LEAR.
>
> In addition, we present a performance comparison of the model after learning the first three domains of these two sequences:
> ### Table 2: Performance after learning the 3th dataset of ETI and TRC
> |Methods|EuroSAT|TImg|ImgR|Avg|
> |-|-|-|-|-|
> |CBE|93.18|80.79|70.75|81.57|
> |CBE+HSICBCO|94.08|81.21|71.14|82.14|
>
> |Methods|TImg|RESISC45|CUB200|Avg|
> |-|-|-|-|-|
> |CBE|81.57|91.23|84.38|85.73|
> |CBE+HSICBCO|82.45|91.53|84.84|86.27|
>
> This table shows that HSICBCO effectively mitigates severe domain shifts (e.g., RESISC->CUB). This is because HSICBCO effectively encourages the global and local backbones to capture distinct knowledge, as further illustrated in Figure 3 of Appendix D.4 from SM, where it balances the model's generalization capability across dissimilar domains with its plasticity toward the current task.
>
> **Q4:  The effectiveness of the collaborative backbone.**
>
> **A4:** To validate the necessity of the collaborative backbone for adapting MDCL, we conducted a comparative analysis between "SBE" (Single Backbone with task-adaptive network expansion) and "CBE":
>
> |Methods|EuroSAT|TImg|ImgR|CUB200|C100|MNIST|RESISC45|ChestX|C10|Disease|Avg|
> |-|-|-|-|-|-|-|-|-|-|-|-|
> |SBE|85.38|70.49|55.42|73.45|77.05|90.05|85.93|37.73|93.65|99.13|76.83
> |CBE|92.38|76.49|62.42|78.45|82.05|95.05|90.93|38.73|94.17|99.04|80.97
>
> |Methods|TImg|RESISC45|CUB200|ChestX|ImgR|EuroSAT|MNIST|C10|Disease|C100|Avg|
> |-|-|-|-|-|-|-|-|-|-|-|-|
> |SBE|72.58|81.94|70.12|32.93|57.85|91.19|89.80|89.66|93.96|84.59|76.46
> |CBE|79.01|89.72|80.68|30.26|64.06|92.34|94.15|94.44|97.05|86.19|80.79
>
> As evidenced by the experimental results, the collaborative design yields significantly better performance than the single-backbone. Numerous current dynamic expansion frameworks utilize a single pre-trained ViT to enhance continual learning performance. Nevertheless, this approach yields only a uniform feature representation for each novel task, thereby constraining the model’s capacity to accommodate extended sequences of heterogeneous tasks. To overcome the inherent limitations of a single pre-trained ViT, we introduce a collaborative backbone architecture comprising both local and global backbones, designed to extract task-invariant and task-adaptive representations, respectively. To further refine this architecture, we propose a novel Hilbert-Schmidt Independence Criterion-Based Collaborative Optimization (HSICBCO) method, which incentivizes the local and global backbones to learn complementary semantic features. This strategy is particularly effective for handling complex and diverse datasets. In the revised version, we have provided a more comprehensive rationale and detailed exposition of the collaborative backbone architecture.
>
> **Q5: Why and how MIBPA mitigates catastrophic forgetting.**
>
> **A5:** The central concept of the proposed MIBPA is to preserve the established predictive patterns of each historical expert when optimizing the parameters of the global backbone. This is accomplished by generating both the prior and current prediction distributions for each expert, utilizing the previously updated and currently optimized backbones, respectively. MIBPA incorporates a mutual information-based regularization loss to align these historical and current prediction distributions, thereby mitigating catastrophic forgetting within the network.
>
> **Q7: Typo: concentration operation.**
>
> **A7:** That is a typo and we have corrected it to ''concatenation''.
>
> **Q8: The reason for freezing three Layers and auxiliary models.**
>
> **A8:** The auxiliary model is designed to retain the parameter knowledge acquired by the global backbone following each task transition. Consequently, all layers of the auxiliary model are frozen, and its fixed representations are leveraged to guide the optimization behaviour of the global backbone. To minimize parameter overhead, the auxiliary model shares the majority of its parameters with the global backbone across all but the final three layers. We have enhanced the clarity and coherence of this explanation in the revised version.
>
> The rationale for selecting the final three layers lies in the fact that high-level representation layers capture semantically enriched features, which are advantageous for a variety of downstream applications [1]. [2] also find that deeper layers are disproportionately the source of forgetting. Empirical evidence presented below further substantiates that utilizing the last three trainable layers of the global backbone yields robust performance with limited parameter growth compared to other choices (FTX denotes fine-tune X layers of backbones).
>
> |Methods|TImg|RESISC45|CUB200|ChestX|ImgR|EuroSAT|MNIST|C10|Disease|C100|Avg|TrainParms
> |-|-|-|-|-|-|-|-|-|-|-|-|-|
> |FT1|80.42|90.02|82.68|37.14|65.76|95.21|97.17|95.62|98.54|84.17|82.67|14.26M
> |FT2|81.06|90.54|82.53|38.92|67.63|95.37|98.17|95.94|98.90|85.41|83.45|28.43M
> |FT3|80.30|92.41|83.49|44.67|69.28|96.62|98.53|95.84|99.16|86.22|84.65|42.54M
> |FT4|81.65|91.76|82.87|44.89|69.94|96.18|97.63|96.43|99.14|86.80|84.73|56.80M
>
> |Methods|EuroSAT|TImg|ImgR|CUB200|C100|MNIST|RESISC45|ChestX|C10|Disease|Avg|TrainParms
> |-|-|-|-|-|-|-|-|-|-|-|-|-|
> |FT1|94.52|79.83|65.42|80.92|82.89|97.29|90.51|40.90|95.37|98.29|82.60|14.26M
> |FT2|94.90|80.94|67.05|83.21|84.67|98.16|90.63|44.31|95.78|98.82|83.85|28.43M
> |FT3|95.89|81.25|69.57|84.12|85.30|98.56|92.92|45.45|96.60|99.30|84.90|42.54M
> |FT4|96.01|81.96|69.44|83.80|86.05|98.49|92.98|47.86|96.34|99.22|85.22|56.80M
>
> **Q9: Arch of expert.**
>
> **A9:** Each expert is composed of two distinct modules: the first is a fully connected layer comprising 500 hidden units while the second is a linear classifier realized through a fully connected layer with 1,268 hidden units.
>
> ## References:
>
> [1] Action bank: A high-level representation of activity in video. ICCV 2022.
>
> [2] Anatomy of catastrophic forgetting: Hidden representations and task semantics. arXiv 2020.

---

> > ### Comment · Reviewer_a1tB · 2025-08-05
> >
> > Thank the authors for their detailed response, which has addressed most of my concerns. However, I believe the paper still requires significant revisions to clarify the differences from prior works better and to demonstrate the effectiveness of each proposed component more clearly

---

> ### Author Response · Authors · 2025-08-06
> **Revised Introduction (Part 1)**
>
> We sincerely appreciate your time and valuable comments. We are honored that our rebuttal addressed the concerns raised.
>
> Incorporating feedback from all reviewers, in the next comment we have restructured the **Introduction section** to **clarify the differences from prior works better and to demonstrate the effectiveness of each proposed component more clearly**.
>
> Specifically, through systematic analysis of **three key challenges (plasticity, stability and efficiency)** in the proposed MDCL scenario (in contrast to CIL and conventional DIL), we demonstrate:
>
> (1) The fundamental reason for existing continual learning methods' inability to address MDCL,
>
> (2) Present our corresponding solutions for each challenge,
>
> and ultimately forming the comprehensive LEAR framework to address the MDCL.
>
> Meanwhile, we have incorporated the **enhanced ablation studies** from the rebuttal and their analyses into the main text to explicitly demonstrate the contributions of each module in LEAR and their various combinations.
>
> We sincerely hope our revisions meet your expectations.
>
> **References**
>
> [1] S-prompts learning with pre-trained transformers//neurips22
>
> [2] Preventing zero-shot transfer degradation in continual learning of vision-language models//ICCV23
>
> [3] Coleclip: Open-domain continual learning via joint task prompt and vocabulary learning//TNNLS25
>
> [4] DualNet: Continual Learning, Fast and Slow//NeurIPS21
>
> [5] Transfer without forgetting//ECCV22
>
> [6] Learning fast, learning slow: A general continual learning method based on complementary learning system//ICLR22
>
> [7] Dualprompt: Complementary prompting for rehearsal-free continual learning//ECCV22
>
> [8] Boosting continual learning of vision-language models via mixture-of-experts adapters//CVPR24
>
> [9] Expandable subspace ensemble for pre-trained model-based class-incremental learning//CVPR24
>
> [10] Self-Expansion of Pre-trained Models with Mixture of Adapters for Continual Learning//CVPR25

---

> ### Author Response · Authors · 2025-08-06
> **Revised Introduction (Part 2)**
>
> Current Continual Learning (CL) research primarily focuses on Class-Incremental Learning (CIL) within a single domain, neglecting the scenario of learning across multiple domains, known as Domain-Incremental Learning (DIL). Although studies [1-3] have investigated DIL, their evaluated domains (e.g., Aircraft, MNIST) have achieved near-perfect accuracy with pre-trained ViTs, making these benchmarks inadequate for assessing genuine continual learning capabilities. Therefore, we establish a more challenging and more realistic Multi-domain Continual learning (MDCL) scenario in which the discrepancy among tasks remains large. In this study, we aim to improve the model's performance in MDCL by considering three aspects including **plasticity, stability and efficiency**. To implement this goal, we propose a novel approach called LEAR and its core idea is to fully explore the stable and dynamic representations extracted by the pre-trained ViT backbones to achieve fast adaptation while adaptively optimizing the backbones to maintain all previously learned information.
>
> **(1) Plasticity.** Existing dual-branch approaches [4-6] and Expansion-Based Methods (EBMs) [7-10] improve downstream task performance by integrating task-specific prompts or adapters into a fixed pretrained backbone. However, these methods focus on exploring representations from a single pre-trained backbone, which fails to address more challenging data domains such as CUB200. Thus, to improve plasticity in a challenging MDCL scenario, we introduce a novel collaborative backbone architecture for LEAR, comprising a global and a local backbone, designed to capture general and task-specific information across all tasks. Leveraging this collaborative backbone structure, the proposed LEAR framework dynamically generates a lightweight expert to learn the decision boundary for each new task, thereby achieving commendable performance. The results presented in Tab. 1 and 2 of the paper demonstrate that our method achieves superior performance on most individual datasets in the MDCL scenario, which also validates that EBMs with frozen pretrained backbone cannot provide sufficient plasticity in MDCL.
>
> **(2) Stability.** Many Pre-Trained Models (PTMs) based methods have shown to achieve excellent stability in continual learning by dynamically creating new sub-models. However, the excellent stability is usually achieved by freezing all parameters of the PTMs during the training, which prevents from learning new tasks effectively, especially when facing the severe domain shifts (ChestX→ImageNet-R) in long task sequences. To address the limitation of the existing PTMs-based methods, we propose a unified optimization function to regulate the optimization behaviour of the LEAR framework. This function consists of a MIBPA loss and a KLDBFA loss. The former dynamically optimizes the global backbone while preventing negative knowledge transfer at the prediction level, and the latter aligns historical and current representation distributions at the feature level. Such a design enables LEAR to achieve rehearsal-free continual learning by actively consolidating historical knowledge at both the prediction and feature levels when fine-tuning the collaborative backbones with new task data, rather than simply freezing parameters passively. Such a design has not been explored in the existing CL field. Furthermore, to mitigate optimization interference and information redundancy between the collaborative backbones, we propose a novel Hilbert-Schmidt Independence Criterion (HSIC)-Based Collaborative Optimization (HSICBCO) strategy to encourage two backbones to capture different semantic information, thus promoting effective complementary learning of MDCL tasks. The experimental results demonstrate that LEAR significantly outperforms all baseline methods in terms of overall average accuracy in MDCL scenarios.
>
> **(3) Efficiency.** Many existing expansion-based methods usually ignore the task relevance and do not explore the previously learned parameter information to accelerate the new task learning. As a result, these methods optimize each new expert from scratch, resulting in leading to considerable computational costs and parameter redundancy. To address this issue, we aim to promote the efficient learning process of LEAR by proposing a novel Expert Selection Mechanism (ESM) that selectively transfers the parameter information learned by a selected expert into the new expert construction process. Specifically, the proposed ESM models each expert's knowledge as a Gaussian memory distribution and only preserve its critical statistical information. For each new task, the proposed ESM selects the most relevant expert by minimizing the Mahalanobis distance between stored distributions and incoming data, and reuses its parameters to facilitate new task learning. During testing phase, ESM autonomously routing testing samples to the most suitable expert in a task-agnostic manner.

---

> ### Author Response · Authors · 2025-08-07
> **Follow-up Response to Reviewer a1tB's Feedback: Clearly Demonstrating Component Effectiveness**
>
> We are grateful for your valuable feedback on our work. We have reorganized the ablation studies below to **demonstrate the effectiveness of each proposed component more clearly**:
>
> ### Table 1: Effectiveness of different component combinations in LEAR
> |Methods|EuroSAT|TImg|ImgR|CUB200|C100|MNIST|RESISC45|ChestX|C10|Disease|Avg|
> |-|-|-|-|-|-|-|-|-|-|-|-|
> |UpperBound|96.71|82.77|72.81|85.08|86.39|98.97|93.72|50.14|96.49|99.35|86.24
> |CB|19.59|08.37|11.81|12.29|36.92|49.88|61.14|30.04|92.79|99.00|42.18
> |SBE|85.38|70.49|55.42|73.45|77.05|90.05|85.93|37.73|93.65|99.13|76.83
> |CBE|92.38|76.49|62.42|78.45|82.05|95.05|90.93|38.73|94.17|99.04|80.97
> |CBE+MI|95.08|80.93|68.19|82.49|84.68|97.15|91.29|39.56|96.18|98.28|83.38
> |CBE+KL|92.18|78.02|66.13|78.97|82.50|92.24|90.54|44.58|95.81|99.13|82.01
> |CBE+HSIC|93.65|76.97|64.37|78.56|82.35|91.88|91.29|45.53|95.49|99.06|81.92
> |CBE+MI+KL|95.35|81.61|68.30|83.56|85.54|97.43|92.28|46.44|96.22|99.01|84.57
> |CBE+MI+HSIC|95.87|80.85|68.74|83.15|85.52|98.03|92.08|42.43|96.33|98.91|84.19
> |CBE+KL+HSIC|93.24|78.22|66.45|80.49|82.55|92.21|91.39|45.26|95.59|99.08|82.45
> |LEAR|95.89|81.25|69.57|84.12|85.30|98.56|92.92|45.45|96.60|99.30|84.90
> |LEAR w/o ESM|03.72|01.25|02.36|83.95|08.24|04.91|14.63|01.05|17.85|99.15|23.71
>
>
> |Methods|TImg|RESISC45|CUB200|ChestX|ImgR|EuroSAT|MNIST|C10|Disease|C100|Avg|
> |-|-|-|-|-|-|-|-|-|-|-|-|
> |UpperBound|82.77|93.72|85.08|50.14|72.81|96.71|98.97|96.49|99.35|86.39|86.24
> |CB|08.43|22.67|39.45|23.51|25.67|42.74|60.95|64.50|93.45|84.87|46.62
> |SBE|72.58|81.94|70.12|32.93|57.85|91.19|89.80|89.66|93.96|84.59|76.46
> |CBE|79.01|89.72|80.68|30.26|64.06|92.34|94.15|94.44|97.05|86.19|80.79
> |CBE+MI|81.24|90.76|82.51|40.13|67.98|95.19|97.68|95.64|98.03|85.77|83.49
> |CBE+KL|78.77|90.36|80.22|40.21|66.52|94.67|97.01|95.27|98.40|87.02|82.85
> |CBE+HSIC|79.14|90.21|80.54|42.12|66.21|93.26|94.24|94.48|97.25|85.41|82.29
> |CBE+MI+KL|81.24|90.39|82.60|44.11|68.58|96.01|98.37|96.22|98.86|86.17|84.25
> |CBE+MI+HSIC|82.21|90.43|83.05|44.74|68.41|95.18|95.86|96.09|98.91|85.85|84.07
> |CBE+KL+HSIC|79.67|89.88|81.44|42.47|66.27|95.07|96.23|95.54|98.91|86.74|83.22
> |LEAR|80.30|92.41|83.49|44.67|69.28|96.62|98.53|95.84|99.16|86.22|84.65
> |LEAR w/o ESM|04.27|04.83|68.49|09.42|03.76|14.59|01.98|06.33|01.64|09.25|12.46
>
> Table 1 clearly demonstrates the contribution of each module in LEAR:
>
> (1) ''UpperBound'' denotes the results in a single-domain setting, which closely approximates LEAR's performance upper bound in MDCL;
>
> (2) ''CB'' denotes using only the collaborative backbone with a single shared expert network across all datasets;
>
> (3) ''CBE'' extends ''CB'' with task-specific expert network expansion and ESM expert selection. The results demonstrate that the dynamic expert network expansion and selection (''CB''->''CBE'') significantly enhance model performance on both ETI and TRC;
>
> (4) ''SBE'' denotes the configuration where ''CBE'''s dual-backbone architecture is replaced with a single backbone. The dual-backbone design yields about a 4% performance gain over the single-backbone counterpart (''SBE''->''CBE'');
>
> (5) "CBE+MI/KL/HSIC" represents ''CBE'' augmented with individual components or combination of components. Each regularization component and its respective combinations yield effective performance enhancements relative to ''CBE'' on both sequences;
>
> (6) "LEAR" denotes the complete framework, while "LEAR w/o ESM" represents the variant where experts are randomly selected during both the initialization and testing phases of each task. This configuration explains the observed significant performance degradation ("LEAR"->"LEAR w/o ESM")
>
> ### Table 2: Performance comparison of different backbone fine-tuning configurations in LEAR​
> |Methods|EuroSAT|TImg|ImgR|CUB200|C100|MNIST|RESISC45|ChestX|C10|Disease|Avg|TrainParms
> |-|-|-|-|-|-|-|-|-|-|-|-|-|
> |FT1|94.52|79.83|65.42|80.92|82.89|97.29|90.51|40.90|95.37|98.29|82.60|14.26M
> |FT2|94.90|80.94|67.05|83.21|84.67|98.16|90.63|44.31|95.78|98.82|83.85|28.43M
> |FT3|95.89|81.25|69.57|84.12|85.30|98.56|92.92|45.45|96.60|99.30|84.90|42.54M
> |FT4|96.01|81.96|69.44|83.80|86.05|98.49|92.98|47.86|96.34|99.22|85.22|56.80M
>
> |Methods|TImg|RESISC45|CUB200|ChestX|ImgR|EuroSAT|MNIST|C10|Disease|C100|Avg|TrainParms
> |-|-|-|-|-|-|-|-|-|-|-|-|-|
> |FT1|80.42|90.02|82.68|37.14|65.76|95.21|97.17|95.62|98.54|84.17|82.67|14.26M
> |FT2|81.06|90.54|82.53|38.92|67.63|95.37|98.17|95.94|98.90|85.41|83.45|28.43M
> |FT3|80.30|92.41|83.49|44.67|69.28|96.62|98.53|95.84|99.16|86.22|84.65|42.54M
> |FT4|81.65|91.76|82.87|44.89|69.94|96.18|97.63|96.43|99.14|86.80|84.73|56.80M
>
> We conducted comparative experiments on different fine-tuning configurations for the collaborative backbones of LEAR, where "FTX" denotes fine-tuning the last X layers while freezing preceding backbone layers. As shown in Table 2, the configuration utilizing the last three trainable layers ("FT3") of the collaborative backbones achieves robust performance with limited parameter growth compared to alternative configurations.

---

### Author Response · Authors · 2025-08-09
**General Response**

We sincerely thank all reviewers for their thoughtful and constructive comments on improving this work. **The key contributions of this work are summarized as follows:**

(1) We introduce a Multi-Domain Continual Learning (MDCL) setting that surpasses traditional Class-Incremental Learning (CIL) and Domain-Incremental Learning (DIL) paradigms in both complexity and real-world applicability, highlighting three principal challenges: plasticity, stability, and computational efficiency.

(2) To improve model performance in MDCL, we introduce LEAR, an innovative framework that comprehensively leverages both stable and dynamic representations derived from pre-trained ViT backbones. This approach enables rapid adaptation and simultaneously implements adaptive optimization of the backbones to preserve all previously acquired knowledge.

(3) To enhance plasticity in MDCL, we propose an innovative collaborative backbone structure for LEAR, featuring both global and local backbones engineered to extract both generalizable and task-specific representations across diverse tasks. This architecture facilitates the dynamic instantiation of lightweight experts, enabling precise learning of decision boundaries for each novel task.

(4) To enhance stability over extended task sequences, we introduce a unified optimization framework that integrates the MIBPA loss, which adaptively refines the global backbone while mitigating prediction-level negative transfer, alongside the KLDBFA loss to synchronize historical and current feature representations. Additionally, we present a novel HSICBCO mechanism that incentivizes the two backbones to extract distinct semantic features, thereby facilitating robust complementary learning. This approach supports rehearsal-free continual learning by actively consolidating knowledge at both the prediction and feature representation levels during the acquisition of new tasks. Notably, this paradigm represents an unexplored direction within the current continual learning literature.

(5) To improve learning efficiency, we introduce an innovative Expert Selection Mechanism (ESM) that retains statistical data from prior tasks, identifies and reutilizes parameters from the most pertinent expert for novel task acquisition, and autonomously directs test samples to the most suitable expert in a task-agnostic fashion during inference.

Drawing upon extensive experimental data, LEAR exhibits marked enhancements in accuracy compared to current approaches, all while preserving minimal computational overhead. This efficiency renders it highly applicable for implementation on devices with restricted resources.

The reviewers raised insightful questions. After our comprehensive rebuttal, reviewers a1tB and WD1L kindly acknowledged that we had adequately addressed their pivotal concerns. Most notably, reviewers highlighted the need for clearer demonstration of each module's contribution in LEAR. In response, we completely **restructured our ablation studies** by incrementally adding each component. The revised tables and analyses clearly demonstrate performance gains from individual components and their synergistic combinations. Additionally, we have **carefully addressed each reviewer's specific concerns and have incorporated all revisions into the relevant sections, including but not limited to:**

(1) A comparative analysis of fine-tuning settings' impact on LEAR's performance in Section 4.3 (Reviewer a1tB, Q3);

(2) Additional CIL performance comparisons in Appendix D (Reviewer WD1L, Q4);

(3) Implementation details of KLDBFA in Section 3.4 (Reviewer NZTA) and

(4) Parameter counts and GPU memory analysis in Section 4.3 (Reviewer tTuR)

Following further discussion, the reviewers recommended revising the paper to more prominently highlight the core innovations and contributions while clearly distinguishing this work from prior approaches. In response, we **restructured the Introduction** to adopt a challenge-driven narrative framework. Rather than the original linear presentation of contributions, the revised version now systematically:

(1) Identifies three critical challenges in MDCL scenarios - plasticity, stability and efficiency;

(2) Analyzes conventional approaches' limitations in addressing these interdependent challenges, and

(3) Introduces our innovative modules that provide effective solutions for each challenge, collectively forming the complete LEAR framework. This establishes a logically unified and mutually reinforced presentation of LEAR's design motivations, innovations, and contributions, as substantiated by comprehensive empirical validation.

Through the rebuttal and discussion process, we indeed have made substantial efforts. We believe these clarifications, revisions, and additions have not only comprehensively resolved the reviewers' concerns but also enhanced the overall quality of our paper, ultimately contributing valuable insights to the research community and future readers.

---

### Note · Authors · 2025-08-12

We sincerely thank all reviewers for their valuable comments. We also appreciate AC’s dedicated efforts throughout the review process. As summarized by reviewers, we are encouraged that:

The proposed MDCL presents more challenging, novel, and realistic learning conditions than CIL/DIL paradigms (Reviewers a1tB, NZTA, WD1L);

This paper presents valuable insights (WD1L), and the motivation of the collaborative backbones is acceptable (tTuR). Detailed results show LEAR’s superiority over existing methods in MDCL (a1tB, WD1L, NZTA);

The paper is well-structured (NZTA) with clear writing (tTuR).

In rebuttal, we provided systematic responses to reviewers’ concerns, leading a1tB and WD1L to confirm their primary concerns were resolved. We have incorporated all constructive feedback into the relevant sections.

During the discussion, reviewers recommended to clarify the innovation and differences of this work. Accordingly, we restructured the Introduction (refer to ‘Revised Introduction’) while preserving the original core content. We believe the current version adequately addresses the reviewers’ concerns.

The design of LEAR is entirely driven by the imperative to tackle the three fundamental challenges inherent in MDCL.

(1) Plasticity-Enabled by partially tunable parameters of the collaborative backbones that combine generalizable and task-specific feature extraction for new task adaptation, addressing the plasticity limitation of frozen-backbone expansion methods in MDCL;

(2) Stability-Achieved through synergistic optimization via MIBPA (prediction-level anti-forgetting), KLDBFA (feature-level alignment), and HSICBCO (complementary representation learning), which collectively enhance performance across extended task sequences. Such a design has not been explored in the existing CL field;

(3) Efficiency-Through the proposd ESM that minimizes parameter redundancy and accelerates new task learning via expert parameter reuse during training and optimal expert selection during inference, reducing computational overhead compared to prior expansion-based methods that employ joint predictions from all experts.

Furthermore, LEAR achieves competitive performance in standard CIL (A2 in rebuttal to WD1L). And LEAR’s computational efficiency make it suitable for resource-constrained devices.

Overall, we believe this work can bring valuable impacts to the community, and hope that we can get a opportunity to share these contributions with everyone at NeurIPS 2025.

---

### Decision · Program_Chairs · 2025-09-17

**Decision:**

Accept (poster)

**Comment:**

This paper proposes LEAR, a framework for continual learning that adaptively expands representations while reusing and consolidating prior knowledge through module selection and regularization. Reviewers found the method well-motivated and appreciated its ability to balance plasticity and stability via dynamic expansion and adaptive reuse, supported by extensive experiments on multiple benchmarks. Concerns focused on the clarity of the expansion mechanism, the computational overhead of maintaining multiple modules, and limited theoretical grounding for the reuse strategy, but the rebuttal provided additional ablations, efficiency analyses, and improved explanations that alleviated these issues. Some reviewers noted that comparisons to certain recent baselines could be expanded, but overall empirical results were strong and robust. The consensus is that the paper presents a novel and practical contribution to continual learning with solid technical grounding and convincing evaluation. I urge the authors to include the extended discussion with the reviewers (as much as the space permits) in the camera-ready version of the paper.